# *Akkermansia muciniphila* as a Next-Generation Probiotic in Modulating Human Metabolic Homeostasis and Disease Progression: A Role Mediated by Gut–Liver–Brain Axes?

**DOI:** 10.3390/ijms24043900

**Published:** 2023-02-15

**Authors:** Huafeng Jian, Yating Liu, Xiaoming Wang, Xinyang Dong, Xiaoting Zou

**Affiliations:** Institute of Feed Science, College of Animal Sciences, Zhejiang University, Key Laboratory of Molecular Animal Nutrition (Zhejiang University), Ministry of Education, Key Laboratory of Animal Feed and Nutrition of Zhejiang Province, Hangzhou 310058, China

**Keywords:** *Akkermansia muciniphila*, biological characteristics, metabolic syndrome, inflammation, aging, neurodegenerative diseases, cancer therapy, gut microbiota

## Abstract

Appreciation of the importance of *Akkermansia muciniphila* is growing, and it is becoming increasingly relevant to identify preventive and/or therapeutic solutions targeting gut–liver–brain axes for multiple diseases via *Akkermansia muciniphila*. In recent years, *Akkermansia muciniphila* and its components such as outer membrane proteins and extracellular vesicles have been known to ameliorate host metabolic health and intestinal homeostasis. However, the impacts of *Akkermansia muciniphila* on host health and disease are complex, as both potentially beneficial and adverse effects are mediated by *Akkermansia muciniphila* and its derivatives, and in some cases, these effects are dependent upon the host physiology microenvironment and the forms, genotypes, and strain sources of *Akkermansia muciniphila*. Therefore, this review aims to summarize the current knowledge of how *Akkermansia muciniphila* interacts with the host and influences host metabolic homeostasis and disease progression. Details of *Akkermansia muciniphila* will be discussed including its biological and genetic characteristics; biological functions including anti-obesity, anti-diabetes, anti-metabolic-syndrome, anti-inflammation, anti-aging, anti-neurodegenerative disease, and anti-cancer therapy functions; and strategies to elevate its abundance. Key events will be referred to in some specific disease states, and this knowledge should facilitate the identification of *Akkermansia muciniphila*-based probiotic therapy targeting multiple diseases via gut–liver–brain axes.

## 1. Introuction

The gut microbiota is a complex ecosystem of microorganisms that inhibits and critically maintains the homeostasis of the gastrointestinal (GI) tract [1]. The microbiota consists of bacteria, archaea, and fungi but also includes their viruses and phages. Probiotics refer to “live microorganisms which when intake in adequate amounts can benefit on host health” as regulated by the FAO/WHO [2,3]. *Akkermansia muciniphila* (hereinafter referred to as Akk) is a strict anaerobe Gram-negative bacterium, first isolated from human feces by Muriel Derrien in 2004, that can use mucin as its sole carbon, nitrogen, and energy source [4]. *Akkermansia muciniphila* and *Akkermansia glycaniphila* (isolated from reticulated python feces) are the only representatives of *Verrucobacterium*, and Akkermansia *muciniphila* is one of the most abundant single species in the human intestinal microbiota (0.5~5% of the total bacteria) [4,5,6]. The host intestinal microbiota is honored as the human second genome and the organ with the greatest digestion. The gut microbiota is the key regulator of host homeostasis and the compositional imbalance of microbial communities commonly causes metabolic disorders and exacerbates disease progression [7]. Since its discovery in 2004, the role of Akk in human metabolic health and disease therapy has been widely studied and therefore it has been called “next-generation beneficial microbes” and/or “one of the most promising probiotics” [8,9,10]. Up to now, more than 566 papers associated with Akk have been published, with obesity and diabetes as the factors under the greatest focus, and cancer has been the hot topic of recent Akk research [11]. In the last decade especially, it was uncovered that Akk can prevent and ameliorate obesity [12], diabetes [13], metabolic syndrome [14], inflammation [15], aging [16,17], negative effects of cancer therapy [18], neurodegenerative diseases [19], etc. However, the exhaustive signaling molecular mechanism of Akk interacting with the host is still not completely understood, although the latest review comprehensively elaborated on the critical functions of Akk in the metabolism of human health and nutrient utilization [20]. As a paradigm for next-generation beneficial microorganisms, Cani and colleagues excellently covered the history of the discovery of Akk and recapitulated most findings and major mechanisms of action of Akk that ameliorate host health (reviewed elsewhere) [21]. Yet, it is worth noting that several studies found that elevated Akk abundance is positively associated with aggravated host disease progression. The aim of this review is thus towards a holistic understanding of the physiological features, biological functions, and strategies for raising the abundance of Akk as well as understanding the molecular mechanism of its interaction with the host. In addition, we also summarize its adverse effects on some special pathological conditions and put forward our views on this.

## 2. The Cultivation Methods, Genome, and Outer Membrane Component Characteristics of *Akkermansia muciniphila*

Akk is strictly anaerobic and mainly grown in mucin medium but also can grow on a limited number of sugars including N-acetylglucosamine, N-acetylgalactosamine, and glucose or a synthetic medium constituted of glucose, N-acetylglucosamine, peptone, and threonine that has been confirmed to be safe for human administration [4,22]. Glucosamine-6-phosphate (GlcN6P) exists in mucin and is also necessary for Akk growth, promoting adaptation to the mucosal niche [23]. In addition, Akk also uses a diverse range of cobamides by changing the cobamide structure via a process termed cobamide remodeling mediated by the CbiR enzyme [24]. In an intestinal hemi-anaerobic coculture system (iHACS), akk uses mucin instead of glucose as a carbon source, as well as when cocultured with human colonic organoids that can metabolize the substrate into acetic acid and propionic acid [25]. Nowadays, the mucin medium is widely and mainly used to cultivate Akk, but the risk of animal-derived mucins is incompatible with human administration. The cultivation and growth of Akk need strictly anaerobic conditions, which make its mass production more challenging, and it still needs further investigations for improving its cultivation method and safety for human oral administration (Table 1).

At present, there are two main sources of Akk strains, which are isolated from the feces of humans and rodents [4,26]. ATCC BAA-835T is the standard strain of Akk, whose genome is composed of one circular chromosome of 2.66 Mbp with an average G+C content of 55.8% [27]. Whole-genome sequencing from 39 Akk isolates derived from human and mouse feces revealed varying genome sizes ranging from 2.65 to 3.20 Mbp (averaging 2.86 Mbp) and almost all (38/39) new isolates were larger than the standard strain, which uncovered the extensive genomic difference of Akk [28]. The reconstruction of 106 newly Akk metagenome-assembled genomes (MAGs) from available metagenomic datasets of human, mouse, and pig gut microbiomes implies a transcontinental distribution of Akk phylogroups across the mammalian gut microbiotas and the acquisition of extra genes, especially antibiotic resistance genes, from symbiotic microbes via recent lateral gene transfer during their mammalian gut evolution history [29]. Genomic and phenotypic analysis of 71 new Akk strains from a cohort of children and adolescents undergoing obesity treatment revealed the genotypic and phenotypic diversity of Akk and some phylogroup-specific traits such as oxygen tolerance, adherence, and sulfur acquisition, which likely influence the colonization of Akk in the GI tract and differentially impact host metabolic and immunological health [30]. Large-scale population genomics analyses of the Akk genus in humans and other animals uncovered a large phylogenetic and functional diversity of the Akk genus in humans [31]. Akk strains in the human gut can be grouped into five distinct candidate species that have remarkable whole-genome divergence despite surprisingly similar 16S rRNA gene sequences [31]. In humans, Akk candidate species display ecological co-exclusion, diversified functional capabilities, and distinct patterns of association with host body mass [31]. Based on CRISPR-Cas loci analysis, new variants and spacers targeting newly discovered putative bacteriophages were found, and increased relative abundance of Akk was observed when cognate-predicted bacteriophages were present [31]. Akk also exhibits subspecies-level genetic stratification with associated functional differences such as a putative exo/lipopolysaccharide operon [31]. The strain differences of Akk trigger differentially anti-inflammatory effects in dextran sodium sulfate (DSS) induced colitis, which may have contributions from the strain’s genomic difference [26]. The functional difference and physiological mechanisms mediated by Akk strain difference are still poorly known to date. Specifically, there is little known about the connection between genome and function, and future research should consider the effects of genome difference on both the functional diversity of multiple genomic backgrounds and strain sources.

In addition to the genome of Akk, the membrane proteins and extracellular vesicles (EVs) of Akk have attracted more attention in the prevention and treatment of obesity and type 2 diabetes (T2D). The first four identified putative neuraminidases annotated in the Akk genome, i.e., Am0705, Am0707, Am1757, and Am2085, cover complementary pH ranges between 4.0 and 9.5, and the temperature optima of the enzymes lie between 37 and 42 °C [32]. The 79 putative outer membrane and membrane-associated extracellular proteins of Akk identified via proteomics include highly abundant proteins involved in secretion and transport, as well as proteins predicted to take part in the formation of pili-like structures, which are observed in Akk [33]. PilQ, a 95-kD protein, is the most abundant outer membrane protein, and is annotated and predicted to be a type IV pili secretin [34]. Interestingly, Amuc-1100, a specific outer membrane protein that is stable at the temperature of pasteurization, has been demonstrated to ameliorate the gut barrier by interacting with toll-like receptor 2 (TLR2) [22]. Amuc_1434*, a recombinant protein Amuc_1434 (expressed in the *Escherichia coli* prokaryote cell system), can inhibit the proliferation of LS174T cells via degrading Muc2 [34,35]. On one hand, Amuc_1434* can block the G0/G1 phase of the cell cycle of LS174T cells and upregulate tumor protein 53 (p53) expression [35]. On the other hand, Amuc_1434* promotes apoptosis, increases mitochondrial ROS levels, and downregulates the mitochondrial membrane potential of LS174T cells [35]. Mechanistically, Amuc_1434* activates the death receptor pathway and mitochondrial pathway of apoptosis by upregulating tumor necrosis-factor-related apoptosis-inducing ligand (TRAIL) and ultimately suppresses LS174T cell viability [35]. Moreover, an 84 kDa purified P9 protein secreted by Akk can facilitate glucagon-like peptide-1 (GLP-1) secretion and brown adipose tissue thermogenesis via interacting with intercellular adhesion molecule 2 (ICAM-2) in an IL-6-dependent manner [36]. However, there is still a lack of sufficient data about the influence of different membrane proteins derived from Akk on human metabolism; further research will be required to fully define and compare the functional divergence by which different membrane proteins regulate host metabolic benefit and human oral administration safety.

Bacterial extracellular vesicles (EVs), submicron-sized bilayer lipid structures, are derived from the cell membrane of both Gram-negative and Gram-positive bacteria and can interact not only with host cells but also other microbes [37]. In vivo and in vitro, EVs have been uncovered to ameliorate obesity and intestinal barrier function in high-fat diet (HFD)-fed mice or Caco-2 cells, which also are decreased in patients with T2D (to be discussed later) [38,39,40]. Recent reports indicate that EVs and live or pasteurized Akk all show a protective role in HFD/carbon tetrachloride (CCl4) induced liver injury [41]. Similarly, EVs are required for the gut microbiotas of children and to cause the Akk-induced bone protective effect [42]. In addition, the genotype differences of Akk also trigger differential metabolic effects on brown adipose tissue inflammation and whitening in HFD-fed mice [43]. This research reveals the beneficial, protective roles of Akk and its EVs in the prevention and therapy of T2D, obesity, and bone loss. Future research will be warranted to know how EVs derived from Akk regulate host metabolism and maintain intestinal homeostasis, as well as evaluate their safe dose and oral administration safety for humans. The above research not only elaborates on the beneficial impacts of Akk and its membrane proteins and EVs on host metabolic health, but also implies its cultivation challenges and that its functional diversity depends on its genome genotype and strain source differences. However, reports regarding the effects of the cultivation medium, genome and/or genotype differences on membrane proteins and EVs derived from different strains of Akk on host homeostasis health are limited (summarized in Table 1). The biological function diversity mediated by genome, genotype, and strain source difference of Akk still needs further investigation before it can enter clinical application due to this question rarely being elaborated on.

**Table 1 ijms-24-03900-t001:** Physiological character and components of *Akkermansia muciniphila*.

Item	Character Description	Function Description	References
*Verrucobacterium*	*Akkermansia muciniphila MucT*	[12,13,14,15,16,17,18,19]	[4,12,13,14,15,16,17,18,19]
	*Akkermansia glycaniphila PytT*	Unknown	[5]
Cultivation medium	Mucin medium: mucin, (Glucosamine-6-phosphate (GlcN6P)), and brain–heart infusion (BHI)	Animal-derived risk	[4,23]
	Limited sugars: N-acetylglucosamine, N, acetylgalactosamine, and glucose	Unknown	[4]
	Synthesis medium: soy peptone (16 g/L), threonine (4 g/L), and a mix of glucose and N-acetylglucosamine (25mM each)	Safe	[22]
Genome component	ATCC BAA-835T: 2.66 Mbp, average 55.8% G+C content	[12,13,14,15,16,17,18,19]	[12,13,14,15,16,17,18,19,27]
	Human/mouse isolated strains: 2.65~3.20 Mbp (averaging 2.85)	[26], largely unknown	[28]
Outer membrane components	Neuraminidases: Am0705, Am0707, Am1757, Am2085; PH:4.0~9.5; Tm:37~42 °C;	Unknown	[22,32,33,34,35,36,37,38,39,40,41,42]
	Extracellular vesicles (EVs)	[37,38,39,40,41,42]	
	Outer membrane proteins: PilQ, a 95-kDa IV pili secretin	Unknown	
	Amuc-1100, stable at pasteurization	[22]	
	Amuc_1434	[34,35]	
	P9--84 kDa	[36]	
Genotypes	Amuc_GP01(Akk I), Amuc_GP25 (Akk II)	[43]	[43]

## 3. *Akkermansia muciniphila*: A Star Probiotic in Preventing and Ameliorating Obesity and Associated Metabolic Disorders

Obesity is a global public health problem in humans. Obesity and its ensuing metabolic disorders, including insulin resistance (IR) and cardiometabolic complications, represent one of the most significant current global health challenges [44]. Obesity and T2D are associated with low-grade inflammation and specific changes in gut microbiota composition such as Akk prevalence [22,44]. Individuals with obesity have a significantly higher abundance of *Firmicutes*, whereas the abundance of Akk is significantly decreased in individuals with obesity and diabetes; it is inversely associated with body fat mass and glucose intolerance [45,46,47]. The abundance of Akk was strongly correlated with body mass index (BMI) and antidiabetic drug usage in Chinese individuals by accurate quantitative analysis [28]. In obese and T2D mice, Akk abundance is significantly decreased and feeding prebiotics can normalize the abundance of Akk, which is correlated with an improved metabolic profile [47]. Akk alleviates HFD-induced metabolic disorders in mice, including fat mass gain, metabolic endotoxemia, adipose tissue inflammation, and IR, as well as increases the intestinal levels of endocannabinoids that regulate inflammation, gut barrier, and gut peptide secretion [47]. Compared with Akk, EVs derived from Akk induce a more significant loss in body and fat weight in HFD-fed mice [39]. Both Akk and EVs ameliorate the intestinal barrier integrity, inflammation, energy balance, and blood parameters (i.e., lipid profile and glucose level) of HFD-fed mice, which implies Akk-derived EVs can be used as a new therapeutic strategy to treat HFD-induced obesity [39]. In vivo, the EVs in the fecal samples of healthy subjects are higher than in patients with T2D [40]. In Caco-2 cells and HFD-induced diabetic mice, EV treatment improves intestinal barrier integrity by increasing tight junction (TJ) protein expression in an AMPK-dependent manner [40]. Lean chow diet-fed mice supplementation with Akk significantly suppresses obesity and associated metabolic profiles and endoplasmic reticulum (ER) stress in the liver and muscle [48]. Akk also ameliorates chronic low-grade inflammation by decreasing plasma levels of lipopolysaccharide (LPS)-binding protein (LBP) and leptin and inactivating LPS/LBP downstream signaling mediated via decreasing JNK phosphorylation and increasing expression of IKBA [48].

A clinical randomized, double-blind, placebo-controlled proof-of-concept and feasibility study in human volunteers who were overweight and obese demonstrated that daily oral administration of 10^9^ or 10^10^ CFU Akk bacteria, either live or pasteurized for three months, was safe and well-tolerated [49]. Oral pasteurized Akk significantly ameliorated insulin sensitivity, decreased insulinemia and plasma total cholesterol, and slightly reduced body weight compared to the placebo group [49]. In addition, oral pasteurized Akk also slightly decreased fat mass and hip circumference compared to baseline [49]. Furthermore, in this IR and metabolic syndrome cohort of overweight or obese subjects, administration of 10^9^ or 10^10^ CFU alive and pasteurized Akk for 12 weeks was followed by shared or specific alterations in the fasting plasma metabolome according to the form of Akk administration compared to the control group [50]. Compared with placebo individuals, overweight or obese subjects administered alive and pasteurized Akk modulated amino acid metabolism by reducing their levels of arginine and alanine and several intermediates of tyrosine, phenylalanine, tryptophan, and glutathione metabolism [50]. Furthermore, a global increase in the levels of acylcarnitine together with the specific modulation of acetoacetate also demonstrated the induction of ketogenesis mediated by enhanced β-oxidation [50]. These results highlight the differential and common alterations of obesity-associated metabolism in individuals with prediabetes after administration of different forms of Akk, such as pasteurized versus alive, and further stress the health-promoting function of Akk. The above two clinical trials indicate that short-term (three months) oral administration of alive or pasteurized Akk is safe, which provides a reference for the clinical application of Akk in human anti-obesity treatment, although we still need more clinical investigations to evaluate the long-term safety of Akk in humans (Figure 1).

Organoids, a novel ex vivo model, can be used to characterize the effect of the microbiota on host epithelium [51]. Akk and its metabolite propionate produced by co-cultivated Akk and organoids affect cellular lipid metabolism by modulating the expressions of fasting-induced adipose factor (FIAF), G protein-coupled receptor 43 (GPR43), histone deacetylase (HDACs), and peroxisome proliferator-activated receptor gamma (PPARγ) [51]. In addition to the live bacteria of Akk and its metabolites, the pasteurization of Akk also equally reduces fat mass development, IR, and dyslipidemia in mice compared with the unpasteurized bacterium [22]. Interestingly, Amuc_1100, a specific protein isolated from the outer membrane of Akk, is stable at pasteurization temperature and improves the gut barrier and partly recapitulates the beneficial effects of Akk by interacting with TLR2 [22]. The genotype of Akk also differentially affects brown adipose tissue inflammation and whitening in HFD-fed mice [43]. HFD mice were administered 5 × 10^9^ CFU Akk for 16 weeks to ameliorate impaired glucose tolerance, hyperlipidemia, and liver steatosis, and the effectiveness of Akk I (Amuc_GP01) was found to be stronger than Akk II (Amuc_GP25) [43]. Compared with Amuc_GP25, Amuc_GP01 intervention significantly inhibited brown adipose tissue whitening and inflammation induced by HFD via repairing the intestinal barrier and relieving endotoxemia [43]. Although Amuc_GP25 has a stronger effect in normal chow diet mice, it does not show the same effects as Amuc_GP01 [43]. The daily oral administration of pasteurized Akk in mice ameliorated diet-induced obesity and decreased food energy efficiency, which may have contributed to an increase in energy expenditure and spontaneous physical activity [52]. Pasteurized Akk not only decreases the contents of perilipin 2 in brown and white adipose tissues but also increases energy excretion in the feces that is not due to the modulation of intestinal lipid absorption or chylomicron synthesis but is instead likely involved in a reduction in carbohydrate absorption and enhanced intestinal epithelial turnover [52]. A recent report found that Akk significantly increases thermogenesis and GLP-1 secretion in HFD-induced C57BL/6J obese mice by induction of uncoupling protein 1 (UCP1) in brown adipose tissue and systemic GLP-1 secretion [36]. Furthermore, an 84 kDa purified P9 protein secreted by Akk alone was sufficient to induce GLP-1 secretion and brown adipose tissue thermogenesis by interacting with ICAM-2, whereas IL-6 deficiency abrogated the effects of P9 on glucose homeostasis and downregulated ICAM-2 expression [36]. Early-life obesity commonly results in neurodevelopmental disorders and defects in learning and memory. Early-life mice fed with HFD exhibited impaired hippocampus-dependent contextual/spatial learning and memory and reduced gut microbiota, especially depleted Akk [53]. Hippocampus-dependent learning and memory deficits reappear in chow-fed mice after transplanting HFD-fed mouse microbiota, whereas the supplementing of Akk can improve gut permeability, decrease hippocampal microgliosis and proinflammatory cytokines (such as TNF-α, IL-1β, and IL-6), and restore neuronal development and synapse plasticity, ultimately ameliorating defects in learning and memory [53]. Chow-fed mice treated with LPS mimicked HFD-induced hippocampus-dependent cognitive impairment, while pharmacological blockade of TLR4 signaling or antibiotics treatment effectively prevented hippocampus-dependent learning and memory deficits in HFD-fed mice [53]. Compared with patients with obesity or asthma, patients with obese asthma usually display additive effects, with higher proinflammatory signatures and changes in the composition of the microbiota, and their asthma disease severity is negatively correlated with fecal Akk level [54]. In the murine model, Akk significantly reduces airway hyperreactivity and airway inflammation, which indicates that Akk may play a non-redundant role in patients with a severe asthma phenotype [54]. These investigations not only highlight the antiobesity benefits of Akk but also stress the functional differences mediated by genotype differences and the anti-neurodegenerative-impairment effect of Akk in early-life obesity induced by a HFD, which provides a novel probiotic therapeutic strategy for the treatment of obesity and neurodegenerative diseases in the central nervous system (CNS) (Figure 1).

Ovariectomy (ovx) commonly results in bone loss, and supplementation with probiotics such as *Lactobacillus rhamnosus GG* can protect from ovx-induced bone loss via producing butyrate in mice [55]. The treatment of Ovx mice with pasteurized Akk can protect them from Ovx-induced fat mass gain but not bone loss, and treatment reduces bone mass in gonadally intact mice [56]. However, in Ovx-induced osteoporotic mice, Liu and colleagues found that colonization with gut microbiotas from children but not the elderly prevented reduction in bone mass and bone strength in conventionally raised mice by reversing the Ovx-induced reduction in Akk [42]. Direct replenishment of Akk is sufficient to prevent Ovx-induced imbalanced bone metabolism and protect against osteoporosis via promoting the secretion of EVs, and this is required to induce child gut microbiota- and Akk-induced bone-protective effects [42]. These nanovesicles can enter and accumulate in bone tissues to attenuate an ovx-induced osteoporotic phenotype by augmenting osteogenic activity and inhibiting osteoclast formation [42]. A possible reason is that the pasteurized Akk fails to reshape the gut microbiotal balance and secrete EVs to drive bone protective effects in an Ovx-induced osteoporotic setting. Above, these reports reveal the benefits of live Akk, pasteurized Akk, and outer membrane proteins and/or EVs derived from Akk in ameliorating obesity and associated low chronic inflammation, intestinal barrier integrity, and cognitive dysfunction during early life in obese humans or murine models. This research also uncovers the bone-protective benefit of EVs derived from alive Akk in Ovx-induced bone loss model mice that does not include pasteurized Akk. These studies in vitro and/or in vivo illustrate that use of Akk may be a potential strategy in anti-obesity, anti-neurodegenerative disorders, and bone loss treatment, although it still needs more clinical trials to evaluate its long-term oral safety (e.g., exceeding three months) in humans before it can be used in clinical treatment (Figure 1).

## 4. *Akkermansia muciniphila*: A New Target for the Therapy of Diabetes?

Diabetes is a global epidemic disease and there will be more than 450 million people worldwide living with diabetes in 2025 [57]. Diabetes is a complex metabolic syndrome characterized by insulin dysfunction and glucose and lipid metabolism abnormalities, including type 1 diabetes (T1D) and T2D [57]. T1D is an autoimmune disorder disease and cumulative evidence has shown that the gut microbiota appears central to the development of T1D and displays a protective role against T1D [58]. The abundance of Akk in the gut microbiota has already been reported to be negatively correlated with T1D in mice and humans, which suggests that it may play a beneficial role against T1D [13]. The colonies with high diabetes incidence exhibited reduced bacterial diversity, AKK in particular, compared with the low-incidence colony in non-obese diabetic (NOD) mice with different diabetes incidence [59]. Oral administration of Akk evokes metabolic and immune signaling to promote immune tolerance in the gut and beyond by promoting mucus production and increasing antimicrobial peptide Reg3γ expression [59]. Akk also inhibits gut *Ruminococcus torque*, decreases serum endotoxin levels, and downregulates TLR expression, eventually promoting regulatory immunity and delaying the development of diabetes [59]. These reports indicate that Akk can ameliorate T1D by activating the immune response and recovering gut microbiota homeostasis. T2D is an obesity-related metabolic disorder disease characterized by hyperglycemia, IR, and insufficient insulin secretion and it is associated with disturbed glucose, lipid, and amino acid metabolism [60,61,62]. Patients with T2D display gut microbiota dysbiosis, including depletion of butyrate-producing bacteria and an increase in various opportunistic pathogens such as *Escherichia* species [63,64]. Compared with lean controls or obesity, patients with T2D display decreased Akk abundance and elevated *Firmicutes–Bacteroidetes* ratio, and GLP1-agonist therapy intervention increased Akk abundance in T2D during the study period [65]. In different diabetic state clinical trials, including prediabetics, newly diagnosed diabetics, diabetics on antidiabetic treatment, and healthy non-diabetics, the abundance of Akk was significantly decreased in treatment-naive diabetics and restored in diabetics on antidiabetic treatment [66]. Antidiabetic treatment recovers the microbial diversity of diabetics in convergence with healthy non-diabetics, which suggests that antidiabetic treatment may contribute to restoring microbial diversity and Akk abundance [66]. Due to the complexity of T2D, more investigations are still warranted to investigate the role of Akk in the development of T2D and its antidiabetic efficacy. Obesity and diabetes and their ensuing series of associated complications are intricate and triggered by many factors. The cross-talk between Akk and host metabolic health is not limited only to the gut but also can exist as a dialogue between multi-organs and/or tissues (i.e., liver, muscle, and adipose tissue) to alleviate obesity and diabetes, and future studies should consider the systemic effect of Akk on the host (Figure 1). In a word, these studies suggest that Akk is a promising candidate probiotic that can be used for obesity and diabetes treatment, although it still needs more clinical trials to test and verify its safety and efficacy.

## 5. *Akkermansia muciniphila*: A Promising Probiotic Candidate for the Prevention and Therapy of Metabolic Syndromes

Obesity, T2D, non-alcoholic fatty liver disease (NAFLD), non-alcoholic steatohepatitis (NASH), and atherosclerosis (AS) are the main representative types of metabolic syndrome and have become some of the major diseases affecting global public health. Here, we will discuss the latest research progress, because there have been excellent reviews about Akk ameliorating metabolic syndrome and/or gastrointestinal diseases in 2019 (reviewed elsewhere) [14,67]. Alcoholic liver disease (ALD), including simple steatosis, fibrosis, and cirrhosis, can deteriorate towards acute alcoholic steatohepatitis (ASH) with high mortality rates, and the present treatment strategies remain limited [68,69]. The abundance of Akk in the fecal matter of patients with ASH is decreased and indirectly correlated with hepatic disease severity compared to healthy controls [69]. Alcoholic hepatitis is one of the most severe types of alcohol-related liver disease. The relative abundance of Akk in the fecal matter of patients with alcoholic hepatitis with more severe disease is significantly decreased [70]. In ethanol feeding of wild-type mice, Akk abundance is decreased, but administration of Akk reverses intestinal Akk depletion induced by ethanol and ameliorates hepatic injury, steatosis, and neutrophil infiltration [71]. Akk also protects against ethanol-induced gut leakiness and increases mucus thickness and TJ expression [71]. ALD model mice fed with berberine exhibited ameliorated acute-on-chronic alcoholic hepatic damage from an increase in the size of the granulocytic-myeloid-derived suppressor cell (G-MDSC)-like population via activation of IL-6/STAT3 signaling, and this changed the overall intestinal microbial community, primarily increasing Akk abundance [72]. Above, these results reveal that supplementation with Akk or recovery of its abundance is an effective strategy for the treatment of ALD.

NAFLD is a progressive fatty liver injury that is associated with obesity and lifestyle modifications, and approximately a quarter of patients with NAFLD subsequently develop NASH, which increases the risk of developing liver cirrhosis and hepatocellular carcinoma (HCC) [73,74]. Metabolic diseases such as obesity, overweightness, T2D, and IR are the main contributors to accelerating the development of NAFLD [75]. Unhealthy lifestyles such as the HFD, the WD, sugar-sweetened beverages, refined carbohydrates, fructose, and high caloric intake all can accelerate the development of obesity, T2D, NAFLD, hyperlipidemia, and atherosclerosis [75]. HFD (45% fat diet) usually causes fatty liver compared to a normal diet with 10% fat. Mice treated with 10^8^ to 10^9^ CFU Akk during the test were prevented from having fatty liver by inhibiting serum triglyceride (TG) synthesis and inflammatory factor levels in the liver and rescuing the mice from the reduction in bacterial diversity caused by HFD to restore gut homeostasis [76]. Oral drugs for treating obesity and associated metabolic diseases prevent the development of NFALD by increasing the abundance of Akk, for instance, Bofutsushosan [77,78], Fucoidan [79], and Liraglutide [80]. A high-fat, high-cholesterol (HFHC) diet usually leads to early NASH in rats with gut microbiota dysbiosis and microbial metabolite alterations, which are accompanied by cognitive impairment [81]. Oral Akk (1 × 10^9^ CFU) significantly reverses HFHC-induced cognitive dysfunction such as impaired spatial working memory and novel object recognition and restores brain metabolism [81]. However, Akk treatment just slightly changes the gut bacterial composition and does not induce major rearrangements of the intestinal microbiota, although it partly increases the microbial diversity [81]. This may imply that Akk can integrate gut microbiota with host brain energy metabolism to ameliorate brain cognitive function mediated by the gut–brain axis. HCC frequently arises in the context of chronic liver diseases, alcoholic and non-alcoholic steatohepatitis being the most common causes. *NEMO*^Δhepa^/Nlrp6^−/−^ mice loss of NLRP6 drives liver disease progression towards fibrosis and cancer, exhibiting increased tumor burden and liver-to-body-weight ratio, enhanced liver injury, inflammatory response, and fibrosis [82]. Loss of NLRP6 is associated with intestinal dysbiosis such as decreased Akk abundance and barrier impairment, including disruption of the intestinal TJ barrier and increased inflammatory gene expression [82]. Intestinal dysbiosis induced by loss of NLRP6 drives the expansion of hepatic monocytic myeloid-derived suppressor cells (mMDSC) and suppression of T-cell abundance in a TLR4-dependent manner, and the reduction in Akk is correlated with mMDSC abundance [82]. Under NLRP6 deficiency, continuous Akk supplementation restores intestinal barrier function and strongly reduces liver injury, inflammation, and fibrosis [82]. Bacterial translocation induced by intestinal dysbiosis triggers increased bacterial abundance in the hepatic tissue of cirrhosis patients, which induces pronounced transcriptional changes, including activation of fibro-inflammatory pathways as well as circuits mediating cancer immunosuppression [82]. This work suggests that the gut microbiota, in particularly Akk, closely shapes the hepatic inflammatory microenvironment, shaping an approach for metabolic-disease-associated cancer prevention and therapy.

Atherosclerosis (AS) is the main contributor to cardiovascular mortality [83]. Metabolomics analysis indicated that the gut microbiota producing proatherogenic trimethylamine-N-oxide (TMAO) after being fed phosphatidylcholine can accelerate AS in mice [84]. After being fed with a high-cholesterol diet, germ-free atherogenic mice with knocked-out apolipoprotein E (Apoe^−/−^) exhibited worsening atherosclerotic lesions compared with conventionally raised mice [85]. Similarly, Apoe^−/−^ mice fed with WD also indicated aggravated development of atherosclerotic lesions that was accompanied by reduction in Akk [85]. Daily oral Akk (2 × 10^9^ CFU) restored the abundance of Akk and significantly ameliorated the atherosclerotic lesions of Apoe^−/−^ mice by alleviating metabolic-endotoxemia-induced inflammation through the restoration of the gut barrier [85]. AS is also an inflammatory disease of the vessel wall, and the major trigger for vessel wall inflammation is hyperlipidemia [86,87]. Katiraei and colleagues found that treatment of hyperlipidemic APOE*3-Leiden (E3L) CETP model mice with 2 × 10^8^ CFU of Akk for 4 weeks was followed by significantly decreased body weight and plasma total cholesterol and TG levels [88]. Supplementation of Akk not only affects the immune cell composition in mesenteric lymph nodes (MLNs) but also reduces the number of total T cells and neutrophils [88]. In addition, Akk also decreases the expression of the activation markers major histocompatibility complex class II (MHCII) on dendritic cells (DCs) and CD86 on B cells and increases the IL-10 production stimulated by LPS in whole blood ex vivo [88]. The colonization of Akk also protected cAMP-responsive binding protein H (CREBH)-null mice from acute and chronic hyperlipidemia by enhancing low-density lipoprotein receptor expression to clear TG-rich lipoprotein remnants, chylomicron remnants, and intermediate-density lipoproteins, as well as alleviating hepatic ER stress and inflammatory response [89]. These results demonstrate that treatment with Akk shows an outstanding anti-atherogenic potential by ameliorating lipid metabolism and activating the immune response to repress the formation of atherosclerotic lesions and inflammation deterioration. Moreover, a recent report found that Akk could prevent cold-related atrial fibrillation (AF) in rats by modulating TMAO-induced cardiac pyroptosis [90]. As one of the most important risk factors for AF, cold exposure is closely related to the poor prognosis of patients with AF, and exposure to cold led to elevated susceptibility to AF and reduction of Akk abundance in rats [90]. Correspondingly, progressively increased plasma TMAO levels were observed in human subjects during cold weather, and the reduction of Akk abundance was an independent risk factor for cold-related AF in a cross-sectional clinical study of human subjects [90]. Oral administration of Akk ameliorates the proAF property induced by cold exposure and restores Akk abundance, which reduces TMAO levels through modulation of the microbial enzymes involved in trimethylamine (TMA) synthesis [90]. However, elevated TMAO can drive the infiltration of M1 macrophages in the atria and increase the expression of caspase1-p20 and cleaved GSdmd, ultimately causing atrial structural remodeling [90]. Furthermore, mice with a conditional knockout of caspase1 can be resistant to cold-related AF. These results uncover the novel causal role of gut microbiota such as Akk and metabolites (i.e., TMAO) in the pathogenesis of cardiovascular diseases, which suggests the possibility of selective supplementation with Akk as a potential therapeutic strategy for the treatment of cardiovascular diseases by targeting the gut–heart axis. Metabolic syndromes, for instance, ALD, NAFLD, NASH, and atherosclerosis, have received attention regarding their development and prevention. Above, these results suggest that treatment with Akk or increasing its abundance displays positive effects on metabolic syndrome and associated metabolic disorders by regulating host lipid metabolism, restoring intestinal microbiota homeostasis, and activating the immune response (Figure 2). In a word, oral Akk or increasing its abundance via diet may be a promising strategy for the prevention and treatment of metabolic syndrome. As described in the previous chapter, the membrane proteins and EVs of Akk show potential anti-obesity and anti-diabetes benefits, and future research should consider the effects of the outer membrane proteins, EVs, and different strains of Akk on metabolic syndrome.

## 6. *Akkermansia muciniphila*: An Anti-inflammatory and Pro-inflammatory Probiotic

Chronic intestinal inflammation usually leads to gut barrier damage and gut microbiota disorder, and it is sensitive to pathogenic bacteria. Inflammatory bowel diseases (IBDs), including Crohn’s disease (CD) and ulcerative colitis (UC) [91], irritated bowel syndrome (IBS) [92], colitis [26], obesity- and diabetes-associated chronic inflammation [93], osteoarthritis (OA) [94], and arthritis [95] are the main types of chronic inflammation. The gut microbiota plays a critical role in IBD’s pathogenesis, progression, and disease severity and is relevant to mucosal immunity and host metabolism [96,97]. Accumulative evidence has shown that the abundance of Akk is decreased in patients with IBD and animal models with colitis, while administration of Akk shows potential anti-inflammatory effects by activating related signaling pathways and immune cells [26,98]. The abundance of Akk is significantly decreased in patients with CD with disease onset below 16 years of age, but there is no difference across different disease states such as healthy controls, UC, IBS, and colorectal cancer (CRC), which implies Akk may be a potential biomarker to assist in pediatric CD diagnosis [98]. Patients with IBD and mice with colitis or colitis-associated colorectal cancer (CAC) display a significant reduction in Akk compared to the healthy control group [99,100]. Treatment with Akk or Amuc-1100 ameliorates mouse colitis by reducing infiltrating macrophages and CD8^+^ cytotoxic T lymphocytes (CTLs) in the colon as well as CD16/32^+^ macrophages in the spleen and MLNs [100]. In addition, Akk also slows down tumorigenesis by expanding CTLs in the colon and MLNs, which triggers TNF-α reduction and PD-1 downregulation by activating CTLs in the MLNs [100].

Mice oral DSS significantly reduces the fecal composition of EVs from Akk and *Bacteroides acidifaciens,* and oral EVs from Akk can protect mice from DSS-induced IBD phenotypes [38]. In vitro, EVs derived from Akk pretreatment colon epithelial cells inhibit IL-6 production induced by EVs derived from *Escherichia coli* [38]. As previously mentioned, Akk improves metabolic-endotoxemia-induced inflammation and alleviates the lean chow diet that triggers mice chronic low-grade inflammation [40,85]. Similarly, oral alive or pasteurized Akk also ameliorates diabetic rat liver injury, gluco/lipotoxicity, oxidative stress, inflammation, and intestine microbiota disorders induced by streptozotocin [101]. Compared with WD, a high-cellulose diet (HCD) is considered an effective method to improve obesity or T2D associated with chronic inflammation. However, oral DSS of mice fed a low-cellulose diet (LCD) drives more severe crypt atrophy and goblet cell depletion, exacerbates gut inflammation, and decreases Akk abundance compared to HCD [102]. Mice fed with HCD or LCD treated with Akk all had crypt length, goblet cell induction, and colitis [102]. Akk, on top of ameliorating mucosal inflammation caused by DSS via the microbe–host interaction by enhancing gut barrier function, reduces the levels of inflammatory cytokines and elevates the diversity of the microbial community [97]. For instance, stable gastrointestinal tract colonization of Akk contributed to the therapy and prognosis of inflammatory intestinal diseases induced by DSS and intestinal radiation via changing the host gut microbial community structure, facilitating proliferation and reprogramming the gene expression profile of *Lactobacillus murinus* [103]. Similarly, cadmium usually induces intestinal damage, and treatment with Akk effectively ameliorates intestinal mucosal damage by producing melatonin [104]. Two strains of Akk were isolated from the feces of humans (designated ATCC BAA-835T) and mice (designated 139), showing differentially anti-inflammatory capacities [26]. In vitro, the two strains of Akk exerted similar anti-inflammatory properties by reducing IL-8 production in TNF-α-stimulated HT-29 cells, whereas neither of them failed to induce the differentiation of regulatory T-cells (Tregs) from the CD4^+^ T cell population significantly [26]. In vivo, both of them ameliorated chronic colitis induced by DSS via improving clinical parameters and downregulating the expressions of TNF-α and IFN-γ in the colons of mice [26]. However, the differentiation of Tregs and the production of short-chain fatty acids (SCFAs) of strain ATCC BAA-835T was stronger than strain 139, although both of them facilitated the normalization of the gut microbiota in DSS mice [26]. These results not only elaborate on the outstanding anti-inflammatory capacity of Akk, they also emphasize the functional divergence mediated by strain difference of Akk.

Differently from Europeans, the healthy Chinese population has a low presence of intestinal Akk, and patients with CD and UC have significantly lower colonization and abundance of Akk [105]. After washed microbiota transplantation (WMT) treatment, more than half of patients achieved clinical responses and significantly increased colonization rates of Akk compared to pre-WMT, which is closely corrected with the efficacy of WMT for IBD [105]. Compared with healthy people, the fecal level of Akk is decreased in patients with UC, and oral administration of Akk ameliorates DSS-induced acute colitis via enhancing the intestinal mucin barrier and suppressing the inflammatory response mediated by NLRP3 activation [106]. *Allobaculum* isolates from patients with UC exacerbate mouse colitis, and it is inversely associated with Akk in human-microbiota-associated mice and human cohorts [107]. However, co-colonization with Akk significantly ameliorates *Allobaculum*-induced intestinal epithelial cell activation and colitis in mice, whereas *Allobaculum* blunts the Akk-specific systemic antibody response and reprograms the immunological milieu in MLNs by blocking Akk-induced DC activation and T cell expansion, which reveals the nonlinear impacts of interspecies interactions between Akk and other microbes on host immunity [107]. The abundance of Akk is decreased in the fecal matter and TLR4 gene expression is upregulated in the intestinal epithelia of patients with UC; these are negatively related to colitis risk, which implies that TLR4 may participate in the development of UC [108]. TLR4 knockout exacerbated DSS-induced colitis of TLR4^−/−^ mice mediated by gut microbiota and the abundance of Akk was significantly decreased in TLR4^−/−^ mice, whereas it was enriched in wild-type mice [108]. FMT and co-housing experiments reveal that predisposing gut microbiota is required for the enhanced susceptibility to colitis of TLR4^−/−^ mice, and TLR4 knockout suppresses RORγt^+^ Treg cells to exacerbate colitis in a gut-microbiota-dependent manner; additionally, colonic RORγt^+^ Treg cell frequency is negatively correlated with colitis severity [108]. Oral administration of Akk efficiently raises the frequency of colonic RORγt^+^ Treg cells to enhance the immune response, and the colonization of Akk in the gut is mediated by the interaction between TLR4 and Amuc-1100 [108]. Recently, Bae and colleagues demonstrated that a lipid derived from Akk’s cell membrane recapitulated the immunomodulatory activity of Akk [109]. This lipid is a diacyl phosphatidylethanolamine with two branched chains (a15:0-i15:0 PE) that can induce the production of TNF-α and IL-6 in DCs [109]. Structure-activity analysis found that the immunogenic activity of a15:0-i15:0 PE depends on its structure-activity relationship and requires its binding to TLR2 and TLR1 to form a TLR2-TLR1 heterodimer [109]. It is worth noting that a15:0-i15:0 PE is less able to induce cytokines than TLR2 agonists such as LPS, and it only induces specific proinflammatory cytokines (e.g., TNF-α and IL-6) [109]. At low doses (1% of EC50), it resets the activation threshold of DCs and reduces the response to subsequent immune stimuli such as LPS, which may partially explain the immunomodulatory effects of Akk [109]. This work stresses that the immunomodulatory activity of Akk is not only limited to Akk itself, but also partially mediated by its outer membrane components, including a15:0-i15:0 PE and Amuc-1100 (as discussed previously) [22]. The antigens from Akk are able to reprogram naïve CD4^+^ T cells into the Tregs lineage, expand preexisting microbe-specific Tregs, and limit wasting disease in the CD4^+^ T cell transfer model of colitis [110]. However, a recent study reported that Akk-induced intestinal adaptive T cell immune responses were only limited to T follicular helper (T_FH_) cells in a gnotobiotic setting during homeostasis [111]. The specific immune response induced by Akk appears in conjunction with robust anti-commensal T-cell-dependent immunoglobulin G1 (IgG1) and IgA but does not appreciably induce T_reg_, T_H_1, T_H_2, or T_H_17 [111]. These findings highlight that Akk-specific immune responses are context dependent and suggest that contextual signals during homeostasis influence T cell responses to the microbiota and modulate host immune function. A recent review elaborated on Akk as a sentinel for gut permeability and its relevance to HIV-related inflammation, and the author suggested that gut microbiota enriched in Akk could reduce microbial translocation and inflammation, lower the risk of developing non-AIDS comorbidities, and improve quality of life in people living with HIV (PLWH) [112]. The relative abundance of Akk was increased in severe fever with thrombocytopenia syndrome virus (SFTSV) infection and reduced in samples from deceased patients [113]. Supplementation with Akk could protect against SFTSV infection by suppressing NF-κB-mediated systemic inflammation via producing β-carboline alkaloid harmaline, which can specifically enhance bile acid-CoA: amino acid N-acyltransferase expression in hepatic cells to increase conjugated primary bile acids, glycochenodeoxycholic acid, and taurochenodeoxycholic acid [113]. Finally, these bile acids induce transmembrane G-protein coupled receptor-5-dependent anti-inflammatory responses to mitigate SFTSV infection [113]. The results of these different studies at present emphasize the immune regulatory function of Akk in the gut by enhancing the immune response mediated by immune cell activation and reshaping intestinal microbiota homeostasis to prevent intestinal inflammation (Figure 2). This research also implies that the beneficial regulatory function of Akk in the gut is linked to the host microenvironment, and future research will be required to fully define the immune response mechanisms by which Akk regulates intestinal inflammation progression. 

*Porphyromonas gingivalis* commonly causes inflammation and periodontal bone destruction, and treatment with Akk suppresses inflammatory cell infiltration and bone destruction in a model of calvarial infection [114]. In vitro, Akk treatment increases anti-inflammatory cytokines and the expression of tight junctional integrity markers but reduces gingipain mRNA expression, which is associated with an increased expression of Akk and pili-like protein Amuc_1100 [114]. Akk or Amuc_1100 ameliorate *porphyromonas-gingivalis*-induced alveolar bone loss by promoting M2 macrophage polarization and improving the production of IL-10 in the gingival tissues of mice, eventually repressing periodontitis [115]. In *porphyromonas-gingivalis*-infected lean and obese experimental periodontitis mice, alive and pasteurized Akk or Amuc_1100 all significantly prevented *porphyromonas-gingivalis*-induced periodontal destruction and inflammatory infiltration while inhibiting TNF-α levels and increasing IL-10 production, which was further confirmed in *porphyromonas-gingivalis*-infected macrophages [115]. In addition, Akk and *porphyromonas gingivalis* co-cultivation significantly increased Akk’s monobactam-related antibiotics synthesis gene expression levels while decreasing *porphyromonas gingivalis’s* gingipain and type IX secretion system gene expression levels [116]. In an acute liver injury mouse model induced by concanavalin A, Akk not only alleviated inflammation and hepatocellular death, but also increased the diversity and richness of the microbial community and simultaneously modulated the variety of gut microbes, the reshaped microbiota contributing to reducing inflammatory cytokines and cytotoxic factors [117]. Similarly, alive and pasteurized Akk or its EVs all exhibit protective effects against HFD and CCl4-induced liver injury by inhibiting hepatic stellate cell (HSC) activation, and EVs display the greatest HSC activity repression [41]. In an excess-acetaminophen-induced liver injury mouse model, administration of Akk efficiently alleviated acetaminophen-induced liver injury by attenuating oxidative stress and inflammation in the liver, this hepatoprotective effect accompanied by the activation of the PI3K/Akt pathway and mediated by regulation of the composition and metabolic function of the intestinal microbiota [118]. For instance, Akk can maintain gut barrier function, reshape the perturbed microbial community, and promote SCFA secretion [41]. These studies reveal that the anti-inflammatory activity of Akk is not only limited to the gut but also contributes to ameliorating extraintestinal inflammation. Together with the beneficial function of Akk in the gut, this jointly highlights that Akk has widely outstanding anti-inflammatory properties in different inflammation-related diseases, which may submit it for consideration as an anti-inflammatory therapeutic agent (Figure 2). Yet, these examples illustrate that the anti-inflammatory function of Akk has been exclusively tested in mice rather than in humans, and thus, future research is warranted to evaluate the treatment impact of Akk on different inflammations including IBD in humans.

However, previously accumulated evidence also shows that Akk can act as a pathobiont to promote colitis and exacerbate the development of inflammation. For instance, Akk exacerbates the inflammatory response caused by murine *Salmonella enterica Typhimurium* and DSS [38,119]. IL-10-deficient mice (IL-10^−/−^) usually display chronic colitis in histology that is similar to humans with IBD [120] and while the IL-10^−/−^ mice treatment with Akk does not promote the development of short-term intestinal inflammation, long-term supplementation of Akk has not been completely evaluated [121]. In both specific pathogen-free and germ-free IL-10^−/−^ mice, the colonization of Akk mediated by NLRP6 was enough to promote intestinal inflammation by inhibiting IL-18 production after modulating the abundance of Akk [122,123]. Akk is partly increased in children with enthesitis-related arthritis (ERA), and the administration of fecal matter from the KRN/B6 x-NOD (K/BxN) model of the RA mouse treated with added Akk to human patients with ERA slightly increased ankle swelling arthritis [38,124]. The potential of Akk in the treatment of IBD has been well elaborated on in a recent outstanding review [125]. Human studies and some mouse studies reveal that the development of IBD and the abundance of Akk are generally negatively correlated, whereas the content of Akk is significantly increased in some mouse models with colonic inflammation that may be associated with the different mouse models used [125]. Adherent-invasive *Escherichia coli* (AIEC) is a pathobiont enriched in the gut mucosa of patients with IBD that utilizes diet-derived L-serine to adapt to the inflamed gut [126]. L-serine metabolism is disturbed in the gut microbiota of patients with IBD, and the deprivation of dietary L-serine exacerbates DSS-induced colitis and leads to blooms of pathotype *Escherichia coli* in the inflamed gut, which are accompanied by a sustained increase in Akk abundance [126]. Further analysis uncovered that Akk enables AIEC to relocate to the epithelial niche by degrading the mucus layer, and AIEC and Akk can cooperate to promote gut inflammation during dietary L-serine restriction [126]. In the epithelial niche, AIEC acquires L-serine from the host colonic epithelium to counteract dietary L-serine deprivation and thus proliferates [126]. This work further highlights the adverse impact of Akk on IBD, enabling pathobionts such as AIEC to overcome nutrition restrictions via exploiting host-derived nutrition mediated by the degradation of the mucus layer by Akk and thus thriving in the gut. Direct Akk intervention treatment suggests that Akk may be a potential probiotic for ameliorating intestinal inflammation, but adverse effects may also occur under certain conditions as mentioned above [38,119,120,121,122,123,124,125,126]. These adverse effects and contradictory results of Akk in the treatment of intestinal inflammation perhaps contribute to the bacterial strain specificity, host pathogenic microenvironment, and mouse strain differences (Figure 2). Therefore, future studies should consider the differences in strain, specific host conditions, and mouse strain when evaluating the therapeutic effects of Akk in the treatment of human inflammation including IBD and colitis.

## 7. *Akkermansia muciniphila* Ameliorates Aging and Aging-Associated Chronic Complications

Aging is a major risk factor for the development of many diseases including cancer [127], chronic inflammation [128], GI disorder [129], and metabolic diseases [130]. Cancer and chronic inflammation incidence substantially increase with aging in both men and women and will become the main cause of reduced life expectancy in the elderly population in the future [131,132]. Gut dysbiosis has been confirmed to play an important role in accelerating the development of aging and chronic inflammation and results in impairment of gut barrier function [126,129,130,131,132]. Aging is commonly associated with chronic inflammation, which also is the main hallmark and risk factor of hyperglycemia and IR with the development of age [128]. Hyperglycemia and IR increasing with age in elderly humans are associated with the accumulation of major histocompatibility complex class I cells (MHCI, also known as 4BL cells) and the loss of the commensal bacterium Akk [131]. IR is induced by activation of innate 4-1BBL^+^ B1a cells, which accumulate in aging with the change of gut commensals and loss of beneficial metabolites (e.g., butyrate) [131]. Furthermore, CCR2^+^ monocytes are activated by endotoxin when butyrate is decreased due to the loss of Akk. Mechanistically, CCR2^+^ monocytes can convert B1a cells into 4BL cells after they infiltrate into the omentum and induce IR by expressing 4-1BBL receptor signaling (128). Supplementation with either Akk or the antibiotic enrofloxacin reverses these pathways and IR by increasing Akk abundance, restoring the normal insulin response in aged mice and macaques [131]. Additionally, treatment with butyrate or antibodies that deplete CCR2^+^ monocytes or 4BL cells shows the same effect on IR [131]. The above results suggest that administration of Akk can inhibit the activation of CCR2^+^ monocytes, resulting in B1a cells failing to convert into 4BL cells, which demonstrates that Akk may be a potential target treatment of aging-associated IR (Figure 3).

Ercc1^−/Δ7^ mice, an accelerated aging mouse model, has a similar accelerated aging phenotype compared to normal aging [132]. Ercc1^−/Δ7^mice treatment with Akk prevents the aging-related decline in the thickness of the colonic mucus layer and attenuates inflammation and the immune-related process by affecting B cell migration to increase mature and immature B cell frequencies in the bone marrow [16]. Akk also decreases the frequency of activated CD80^+^CD273-B cells in Peyer’s patches and Ly6C^int^ monocyte frequencies in the spleen and MLNs but increases the number of peritoneal resident macrophages [16]. Premature aging disorders, such as Hutchinson-Gilford progeria syndrome (HGPS, or progeria) and Nestor-Guillermo progeria syndrome (NGPS), are some of the rarest human diseases [133,134]. Consistent with human progeria patients, the abundance of *Proteobacteria* and *Cyanobacteria* are increased but *Verrucomicrobia* is decreased in both Lmna^G609G/G609G^ mice and Zmpste24^−/−^ mouse models, whereas long-lived humans (that is, centenarians) exhibit a substantial increase in *Verrucomicrobia* and *Proteobacteria* reduction [17]. Fecal microbiota transplantation (FMT) from wild-type mice ameliorates healthspan and promotes lifespan in both progeroid mouse models and transplantation with *Verrucomicrobia* Akk is sufficient to exert beneficial effects that are correlated with the restoration of secondary bile acids [17]. Administration in aging mice of *Lactobacillus acidophilus DDS-1* increases the abundances of Akk and *Lactobacillus spp* and reduces *Proteobacteria spp* [135]. Similarly, supplementation with *Lactobacillus paracasei D3-5* and Lipoteichoic acid derived from the cell wall of heat-killed *Lactobacillus paracasei D3-5* extends the life span of *Caenorhabditis elegans* and improves HFD-induced mouse metabolic dysfunctions, gut barrier impairment, inflammation, and physical and cognitive functions and also increases mucin production and Akk abundance [136]. Compared with young mice, the relative abundance of Akk is decreased in aged mice, whereas the gut microbial functions involved in butyrate and γ-aminobutyric acid (GABA) biosynthesis are significantly enriched in young mice [137]. Shin and colleagues revealed that the abundance of Akk is significantly increased in co-housing and serum injection mice (from young to aged), whereas it is dramatically reduced in parabiosis mice (where the two mice are connected through the elbow and knee joints, as well as the skin), and the abundance of Amuc_1100 is also significantly increased in co-housing and serum injection mice with the increase in Akk level, which indicates that co-housing and serum injection stimulates Akk growth in the intestine of aged mice [137]. Aged mice administration of Akk (4.9 × 10^8^ CFU) ameliorates intestinal integrity and homeostasis by inhibiting the infiltration of the proinflammatory molecule LPS, accelerating the renewal and turnover of intestinal cells, and increasing the abundance of Akk in the gut microbiome [137]. Furthermore, oral administration of Akk is non-toxic at given doses and extends the healthy lifespan via elevating the frailty index, cognitive function, and restoration of muscle atrophy [137]. The above results suggest that Akk ameliorates aging and aging-associated metabolic disorders by regulating host immunity and recovering gut microbiota homeostasis mediated by elevating Akk abundance and its metabolites including butyrate, secondary bile acids, and GABA, which suggests that Akk may be a potential aging therapy strategy, although it still warrants more investigations and clinical trials to test this (Figure 3).

## 8. *Akkermansia muciniphila*: A Promising Neurodegenerative Disease Therapeutic Agent

Neurodegenerative diseases, for instance, autism spectrum disorder (ASD) [138], Alzheimer’s disease (AD) [139], Parkinson’s disease (PD) [140], multiple sclerosis (MS) [141], and amyotrophic lateral sclerosis (ALS) [142], are CNS diseases with serious neurological dysfunction and associated complications. Altered gut microbiota has been associated with neurodegenerative diseases, especially the abundance of Akk [19,143,144,145,146]. Children with autism show a lower relative abundance of Akk in fecal matter, which is associated with a change in mucus barrier [143]. Compared to healthy people, patients with ALS display an altered gut microbiota [146]. ALS-prone Sod1 transgenic (Sod1-Tg) mice show pre-symptomatic, vivarium-dependent dysbiosis and altered metabolite configuration, and germ-free conditions or broad-spectrum antibiotics treatment all exacerbate the disease [147]. The symptoms of ALS are associated with Akk, and treatment with it ameliorates the symptoms of ALS mice, while supplementation with *Ruminococcus torque* and *Parabacteroides distasonis* exacerbate ALS [148]. Akk treatment increases nicotinamide in the CNS, and systemic supplementation of nicotinamide also promotes motor symptoms and gene expression patterns in ALS mice, which is further evidenced by reduced levels of nicotinamide systemically and in the cerebrospinal fluid in human ALS patients [148]. This result would be an indicator for the treatment of ALS that Akk or nicotinamide is a candidate therapeutic agent, although it is still unknown how Akk promotes the production of nicotinamide (Figure 3).

As it is an autoimmune disease directly against the CNS, myelin MS is associated with demyelination, oligodendrocyte loss, reactive gliosis, and axonal degeneration [141]. Fecal microRNAs (miRNAs) participate in shaping the host gut microbiome, and oral administration of microRNA-30d-5p (miR-30d) from the feces of patients with MS can suppress MS-like symptoms in mice by increasing Akk abundance [145,149]. FMT feces from the experimental autoimmune encephalomyelitis (EAE) model of MS ameliorated recipient disease progress through a miRNA-dependent manner and increased miR-30d in the feces of peak EAE and untreated patients with MS and expanded Tregs [145]. Mechanistically, orally synthetic miR-30d increases the expression of β-galactosidase and gut abundance of Akk, and, in turn, Akk suppresses EAE symptoms by increasing Tregs [145]. AD is a serious cognitive impairment and generalized dementia disease, and Akk abundance is decreased in AD model mice [138]. Treatment with Akk ameliorates cognitive deficits and amyloid pathology by improving glucose tolerance, intestine barrier dysfunction, dyslipidemia, and pathological changes in the brain and relieving the impairment of spatial learning and memory in AD model mice induced by HFD [150]. However, it is worth noting that Akk was found to be enriched in patients with PD by a clinical meta-analysis and whether Akk would exacerbate the progression of PD still warrants further investigations [140]. These results elaborate on the protective effects of Akk on multiple neurodegenerative diseases, which provides a paradigm for the treatment of CNS neurodegenerative diseases based on probiotic therapy in future medicine, although these beneficial functions were exclusively tested in mice and a giant interspecies challenge between rodents and humans still exists (Figure 3).

## 9. *Akkermansia muciniphila* Influences Efficacy of Cancer Prevention and Therapy

Akk is abundantly present in healthy humans, and recent several reports have found that Akk can influence the efficacy of cancer therapy. As previously mentioned, the relative abundance of Akk is reduced in HCC model mice, and continuous treatment with Akk restores the diversity of gut microbiota and reduces liver injury, inflammation, and fibrosis [82]. Eradication of *Helicobacter typhlonius* in young conventional mice using antibiotics decreased the number of intestinal tumors, and the additional presence of Akk prior to antibiotic treatment reduced the tumor number even further [18]. Colonization of pathogen-low intestine-specific conditional Apc mutant mice (FabplCre; Apc^15lox/+^) with *H.typhlonius* or Akk increased the number of intestinal tumors and the thickness of the intestinal mucus layer, and Akk colonization without *H.typhlonius* increased the density of mucin-producing goblet cells [18]. However, dual colonization with *H.typhlonius* and Akk significantly reduced the number of intestinal tumors, mucus layer thickness, and goblet cell density to that of control mice [18]. Further, global microbiota composition analysis found a positive association of Akk and *H.typhlonius* with increased tumor burden, which suggests that the tumor suppression effect of Akk depends on the host intestinal microenvironment and can interact with gut microbes [18]. Immune checkpoint inhibitors (ICIs) targeting the PD-1/PD-L1 axis induce a sustained clinical response in a sizable minority of patients with cancer and are correlated with intestinal Akk abundance. For instance, fecal Akk is associated with the clinical benefit of ICIs in patients with non-small-cell lung cancer (NSCLC) or kidney cancer [151]. FMT from patients with NSCLC who respond to ICIs in germ-free or antibiotic-treated mice ameliorates the antitumor effect of PD-1 blockade, whereas FMT from nonresponding patients fails to do so, which is correlated with the relative abundance of Akk [151]. Oral supplementation with Akk after FMT with non-responder feces restores the efficacy of PD-1 blockade in an IL-12-dependent manner by increasing the recruitment of CCR9^+^CXCR3^+^CD4^+^T lymphocytes into mouse tumor beds [151]. Further shotgun-metagenomics-based microbiome profiling in a large cohort of patients with advanced NSCLC (n = 338) treatment with first- or second-line ICIs revealed that baseline stool Akk is associated with increased objective response rates and overall survival in multivariate analyses, independent of PD-L1 expression, antibiotics, and performance status [152]. Intestinal Akk is accompanied by richer commensalism, including *Eubacterium hallii* and *Bifidobacterium adolescentis*, and a more inflamed tumor microenvironment in a subset of patients [152]. However, antibiotic use (20% of cases) coincided with a relative dominance of Akk above 4.8%, accompanied by the genus *Clostridium*, both associated with resistance to ICIs [152]. This research further demonstrates that the relative abundance of Akk in the gut and feces may represent potential biomarkers to refine patient stratification and contribute to the treatment of NSCLC via ICIs targeting the PD-1/PD-L1 axis. 

Abiraterone acetate (AA) is an inhibitor of androgen biosynthesis for prostate cancer treatment Patients receiving oral AA treatment demonstrate increased Akk abundance, and the altered microbial diversity indices depend on the background Akk level; it has been further evidenced in pure culture that AA uniquely promotes Akk growth [153]. The ketogenic diet (KD) is used to treat refractory epilepsy, and the gut microbiota is changed by KD. Antibiotic treatment or germ-free rearing depletes the KD-mediated seizure protection function in mice, whereas enrichment of, and gnotobiotic co-colonization with, KD-associated Akk and *Parabacteroides* restores seizure protection [154]. FMT from the KD gut microbiota and treatment with Akk and *Parabacteroides* each confer seizure protection to mice fed a control diet, this being associated with alterations in colonic lumen, serum, and hippocampal metabolomic profiles correlated with seizure protection, including reduction in systemic gamma-glutamylated amino acids and elevation of hippocampal GABA/glutamate levels [154]. In vivo, bacterial cross-feeding decreases gamma-glutamyltranspeptidase activity and inhibits gamma-glutamylation to promote the seizure protection function [154]. Akk is decreased in ovarian cancer patients and closely related to ovarian cancer progression, whereas Akk supplementation with FMT significantly suppresses ovarian cancer progression in mice [155]. Akk supplementation with FMT elevates Akk abundance and accompanies acetate accumulation, which is associated with enhanced IFN-γ secretion of CD8^+^ T cells and their tumor-killing properties [155]. In colitis-associated colorectal cancer (CAC), Akk abundance is decreased and supplementation with Akk slows down tumorigenesis by expanding CTLs in the colon and MLNs to trigger TNF-α reduction and PD-1 downregulation [100]. Fan and colleagues found Akk abundance was significantly reduced in patients with colorectal cancer (CRC) [156]. Treatment with Akk suppressed colonic tumorigenesis in *Apc*^Min/+^ mice and the growth of implanted HCT116 or CT26 tumors in nude mice by facilitated enrichment of M1-like macrophages in an NLRP3-dependent manner [156]. They also found that TLR2 was essential for the activation of the NF-kB/NLRP3 pathway and Akk induced M1-like macrophage response; the M1-like macrophage and NLRP3/TLR2 are positively associated with Akk in patients with CRC [156]. These results emphasize the importance of Akk-induced immune activation in cancer treatment by enhancing gut homeostasis and improving the immune microenvironment, providing a therapeutic target in the GI tract cancers including HCC. These current studies highlight the potential benefit of Akk in the treatment of multiple neurodegenerative diseases and cancer via regulating host immune response and metabolism mediated by gut microbiome reshaping (Figure 3). Yet, some of the above reports also imply that Akk may interact with some specific microbes or be affected by the host gut microenvironment, which means we must be cautious about its application in clinical cancer treatment.

## 10. *Akkermansia muciniphila*: Interaction with Intestinal Barrier

The intestinal barrier consists of a mechanical barrier, chemical barrier, immune barrier, and biological barrier, which prevent pathogens and toxins from entering the body through the intestinal mucosa [157,158]. As explained above, Akk can ameliorate gut barrier impairment in the cases of obesity [22,39,40,47], metabolic diseases [71,76,85], inflammation [85,122,123], and aging [125,129,130,131,132]; therefore, it is necessary to know how Akk interacts with the host intestinal barrier. As an essential part of the innate immune system, the mucus barrier is secreted by epithelial goblet cells and its main component is the gel-like properties of polymeric mucins, which not only prevent intestinal epithelium from direct contact with intestinal microbes, but also serve as a substrate for mucus-degrading bacteria, including Akk [159,160]. As one representative of mucus-degrading bacteria, Akk specializes in the utilization of mucin as a carbon and nitrogen source and efficiently colonizes the colon, which produces most of the mucins and is closely associated with the intestinal cells [4,161]. Apart from using mucin as a carbon and nitrogen source, alive Akk also improves the metabolic profile and mucus layer thickness, whereas heat-killed Akk fails to do so [47]. Colonization of Akk changes mucosal gene expression profiles, with increased expression of genes involved in immune response and cell fate determination, in germ-free mice [161]. Akk strongly adheres to Caco-2 and HT-29 human colonic cells and binds to the extracellular matrix protein laminin by improving enterocyte monolayer integrity mediated by increasing transepithelial electrical resistance (TER) and inducing the production of IL-8 [162]. The intestinal mucosa, as an important immune barrier, plays an indispensable role in protecting the integrity of the gut barrier, and intestinal mucosal damage is generally observed during inflammations such as IBD and enteric infections, etc. [91]. Akk stimulates the proliferation and migration of enterocytes adjacent to colonic wounds in a process involving FPR1 and intestinal epithelial-cell-specific NOX1-dependent redox signaling during the damaging of murine mucosa [163]. Colonization of Akk ameliorates intestinal mucosal damage associated with inflammation, gut morphological damage, and body weight of chickens caused by *S. pullorum* infection via accelerating proliferation of intestinal epithelium mediated by the activation of the Wnt/β-catenin signaling pathway and the restoration of the damaged intestinal mucosa, increasing the number of goblet cells, and up-regulating the expression of Muc2 and trefoil factor 2 (Tff2) [164]. The treatment of mice with Akk for 4 weeks significantly accelerated the proliferation of Lgr5^+^ intestinal stem cells (ISCs) mediated by Wnt signaling and promoted the differentiation of Paneth cells and goblet cells in the small intestine (SI) and colon, which was further evidenced by SI organoids cultured with cecal contents obtained from Akk-treated mice being larger and dependent on GPR41/43 [165]. Akk treatment also promotes SCFA secretion and ISC-mediated epithelial development, as well as changing the composition of the gut microbiota and SCFA production, which was evidenced in germ-free mice [165]. Pre-treatment of mice with Akk ameliorates gut damage caused by radiation and methotrexate, and a strain of Akk isolated from healthy human feces is superior to the BAA-385 strain in terms of ISC-mediated epithelial development [165]. These results suggest that Akk strengthens the intestinal barrier via triggering ISC proliferation to promote intestinal epithelium cell development and proliferation and again highlight an Akk strain’s functional difference dependent on the source of the strain (Figure 2). For example, Akk colonization alleviates high-fructose- and restraint-stress-induced jejunal mucosal barrier disruption by enhancing the function of NLRP6, promoting autophagy, maintaining the normal secretion of antimicrobial peptides in Paneth cells, promoting the expression of tight junction proteins, negatively regulating the NF-kB signaling pathway, and inhibiting the expression of inflammatory cytokines [166]. A recent report suggests that Akk upregulates genes involved in maintaining the intestinal barrier function via ADP-heptose-dependent activation of the ALPK1/TIFA pathway, which indicates that Akk promotes intestinal barrier homeostasis by activating an innate immune response [167]. Amuc-1100 promotes IL-10 production by activating TLR2 and TLR4 and eventually activates the NF-kB pathway to promote the production of cytokines [168]. Interestingly, Amuc-1100 also ameliorates obese and diabetic mouse metabolism by interacting with TLR2 [22,168]. EVs are derived from Akk and enhance TJ function by activating AMPK, reducing body weight gain, and improving glucose tolerance in HFD-induced diabetic mice [40]. In vitro, EVs ameliorate gut permeability by interacting with TLR2 and TLR4 and increasing the expression of TJ proteins in Caco-2 cells [37,48]. These results indicate that Akk improves intestinal integrity through promoting intestinal mucin secretion and gut epithelium proliferation and development and inducing host immune cell activation to further strengthen the host immune response, together with maintaining normal intestinal barrier function (Figure 2).

## 11. Strategies to Elevate the Gut Abundance of *Akkermansia muciniphila*

The beneficial functions towards host metabolic health and disease treatment of Akk have been believed to make it the most highly promising probiotic therapeutic agent in future medicine [8,9,10,20,21]. On the one hand, the beneficial impact of Akk on certain kinds of diseases is still emerging, including obesity, diabetes, metabolic syndrome, aging, inflammation, neurodegenerative diseases, cancer, and so on. On the other hand, Akk is a strictly anaerobic Gram-negative strain whose growth needs strictly anaerobic environments in vitro. Yet, the clinical use and evaluation of Akk, especially as a long-term therapeutic supplement, is still limited and unapproved by the FAO/WHO. The current results imply that the dual effects of Akk on rodents and humans are contradictory and warrant further investigations to evaluate its dose, safety, and toxicity in humans. Therefore, how to improve Akk abundance by dietary intervention has become a top priority and is a conservative but effective strategy for the treatment of multiple diseases. Here, we will discuss and supplement recent new evidence because there has been an excellent dissertation about the abundance of Akk resulting from a dietary intervention [20,169,170]. Dietary supplements, probiotics, fructooligosaccharides or FOS (also referred to as prebiotics), FODMAP, polyphenol, metformin, rhubarb (Da Huang), caloric restriction (CR), antibiotics, and lipids all can increase Akk abundance [169]. For instance, oral administration in diet-induced obesity mice of a mixture of 5 × 10^8^ CFU, *Lactobacillus rhamnosus LMG S-28148,* and *Bifidobacterium animalis subsp. lactis LMG P-28149* for 14 weeks significantly increased the abundance of Akk approximately 100-fold in the fecal content [171]. *B. animalis*, not *L. rhamnosus*, was the main contributor, and oral *Bifidobacterium animalis subsp. Lactis 420* significantly increased Akk abundance [171,172]. FODMAP refers to fermentable Oligo-, Di- and Mono-saccharides and Polyols, which include fructose, lactose, oligosaccharides, polyols, and sugar alcohols such as sorbitol, mannitol, xylitol, and maltitol, which due to lack of hydrolases in the small intestine result in limited transport across the epithelium (fructose) and rapid fermentation [173]. Two human clinical intervention studies between patients with IBS and healthy patients found that the abundance of Akk is positively associated with FODMAP in diets in different patients/health subjects [174,175]. Dietary polyphenols such as cranberry extract (CE), black raspberries, pomegranate ellagitannin, epigallocatechin-3-gallate (EGCG), and grape proanthocyanidin significantly ameliorate gut inflammation and related metabolism indicators as well as increases the abundance of Akk in mice [169,176,177,178,179,180,181,182]. However, not all dietary polyphenols can enhance the abundance of Akk, for example, pomegranate extract, green tea extract, and whole California table grapes [169].

Metformin, the most popular first-line drug for the treatment of T2D, can ameliorate gut microbiota composition through an increased abundance of Akk, as well as several SCFA-producing microbiotas [169,183]. Rhubarb (a Chinese herbal medicine), caloric restriction, and vancomycin treatment also promote the abundance of Akk, whereas HFD and alcohol reduce it [169]. Compared with lard, mice fed fish oil for 11 weeks (dietary lipid-enriched omega-3 polyunsaturated fatty acid) demonstrated significantly increased abundance of Akk and *Lactobacillus* in their cecal contents [184]. Resveratrol, polydextrose, yeast fermentation, sodium butyrate, and inulin all increase Akk abundance, while a diet low in fermentable oligosaccharides, disaccharides, monosaccharides, and polyols decreases it [185]. Specific-pathogen-free and germ-free mice intake of or colonization with Akk combined with the intake of a chicken-protein-based diet demonstrated significantly increased Akk abundance and strengthened gut mucus barriers such as an increased number of goblet cells, Muc2 mRNA expression level, and colon mucus layer thickness compared with mice fed a soy-protein-based diet [186]. Berberine (BBR), a Chinese medicine, oral BBR, and its structural analog dihydroberberine can better increase Akk abundance [187]. Underfeeding and oral vancomycin intervention result in greater stool calorie loss; over- and underfeeding have a minor impact on gut microbial community structure, but oral vancomycin significantly changes community structure and reduces diversity, and both interventions increase Akk abundance [188].

However, increased Akk abundance may bring negative effects on host health and even increase the risk of some diseases. For instance, a large cohort study of Chinese people with T2D performed using a metagenome-wide association study (MGWAS) found that Akk was enriched in patients with T2D fecal matter and the abundance of Akk was positively correlated with T2D [63]. Administration in IL10^−/−^ mice of two commonly used food emulsifiers, carboxymethylcellulose (CMC) or polysorbate-80 (P80), both reduced microbial diversity, but, in particular, promoted the abundance of Akk, colitis, and metabolic syndrome, which may have triggered an excessive immune response, thereby disrupting mucus secretion and causing intestinal barrier damage in the IL10^−/−^ mice, demonstrating that Akk is involved in promoting disease progression in some specific situations [189]. These results further again emphasize the adverse impacts of Akk in some specific host microenvironments and that strategies to raise the Akk abundance should consider the host’s physiological and genetic background. Betaine is a trimethylated derivative of glycine. Higher betaine intake during lactation not only increases milk betaine content in dams, leads to lower adiposity, and improves glucose homeostasis throughout adulthood in mouse offspring, but is also accompanied by a transient increase in Akk abundance in the gut during early life and a long-lasting increase in intestinal goblet cell number [190]. Infants exposed to higher milk betaine content during breastfeeding show higher fecal Akk abundance, and mouse pups during the lactation period treated with Akk partially replicated the beneficial effects of maternal breast milk betaine, including increased intestinal goblet cell number, lower adiposity, and improved glucose homeostasis during adulthood [190]. D3, a 9-amino acid peptide, can ameliorate leptin resistance and upregulate uroguanylin (UGN) expression by suppressing appetite via the UGN-GUCY2C endocrine axis, ultimately alleviating diet-induced obesity of mice and macaques [191]. D3 treatment also significantly changes the composition of gut microbiota, specifically increasing the abundance of Akk approximately 100-fold, whereas D3 does not promote the growth of Akk in vitro [191]. Akk transplantation efficiently decreases body weight and lipid metabolism of obese mice in a way that is mediated by the combined effect of D3 and Akk; however, Akk or D3 treatment significantly increases the thickening of the mucus layer, which is what contributes to the increased Akk abundance rather than the direct action of D3 [191]. You and colleagues found *Bacteroides vulgatus* SNUG 40005 could restore the abundance of Akk reduction induced by HFD, whose enrichment was attributed to the metabolite N-acetylglucosamine produced during mucus degradation by *Bacteroides vulgatus* SNUG 40005 [192]. These results suggest that dietary intervention may be a conservative but low-risk strategy to increase the abundance of Akk before it is approved for clinical therapy, although there have been clinical trials confirming that continuously administrating it for three months does not show adverse reactions. However, to be conservative, there must be more large-scale and long-term clinical trials to evaluate the safety of oral administration of Akk.

The large potential of Akk as a novel probiotic therapeutic agent in future medicine has been uncovered in human and animal model studies. Yet, Akk is so strictly anaerobic and extremely sensitive to oxygen that this results in it being difficult to be separated, purified, cultivated, and preserved, and it is limited only to laboratory-scale production and not large-scale industrial production. In addition, at present, the mucin medium derived from animal proteins that is widely used to efficiently cultivate Akk may cause allergic reactions and pathogenic bacteria or viruses to spread in the human body. The last and most important point is that there are currently few experiments or clinical trials to evaluate the toxicology, safety, and optimal amount of Akk added in food or orally administered, and the effect of long-term intake on intestinal homeostasis and host metabolic health urgently needs to be assessed. Dietary intervention thus may be a conservative but efficient strategy to elevate intestinal Akk abundance to promote host health and metabolism based on the host’s physiological environment and genetic background.

## 12. Conclusions and Future Directions

The critical contributions of Akk toward human metabolic health have just begun to be elucidated. In particular, more recent research is revealing how the impacts of Akk extend beyond the GI tract, especially for the so-called gut–brain (e.g., CNS diseases) [133,134,135,136,137], gut–liver (e.g., liver diseases) [14,65], gut–bone (e.g., bone loss) [42], gut–heart (e.g., AS) [81,90], and gut–adipose and gut–muscle tissue (e.g., adipose and muscle) axes [22], as well as cancer therapy [151]. The primary focus to date has been on the use of Akk and its components to ameliorate obesity, diabetes, metabolic diseases, inflammation, neurodegenerative diseases, aging, and the negative effects of cancer therapy, and crucial knowledge gaps remain in this area, specifically on how Akk modulates lipid and glucose metabolism, brain metabolism, and the immune response. However, the safe dose of Akk and its negative effects are comparatively less well understood. Due to both the functional diversity of Akk and the complexity of human microbiome metabolism, this is not merely a matter of such functions affecting health positively or negatively, but rather how they are balanced. Moreover, due to the limited research on Akk strain genome- and genotype-difference-mediated function diversity as well as metabolic difference, the effects of Akk supplementation in the prevention and treatment of obesity, diabetes, inflammation, metabolic diseases, aging, cancer, and neurodegenerative diseases still warrant more intensive work. Future work would therefore aim to determine which of these pathways are upregulated and downregulated in different disease states, such as CNS disease (gut–brain axis), metabolic disease (gut–liver axis), bone metabolism disease (gut–bone axis), cardiovascular disease (gut–heart axis), and cancer treatment. Furthermore, a combination of long-term human clinical administration- and culture-omics (in vitro and in vivo)-based studies would resolve the safety, tolerance, and functional diversity of Akk present among differently diseased individuals in order to further enhance our understanding of Akk colonization in gut microbial ecosystems and thus gain a necessary perspective for improving wellness.

## Figures and Tables

**Figure 1 ijms-24-03900-f001:**
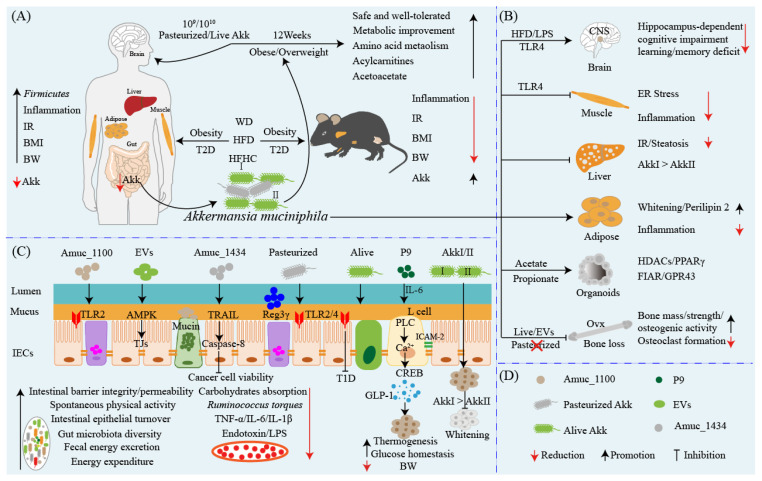
Cross-talk between *Akkermansia muciniphila* and multiple gut organs alleviates obesity and diabetes. (**A**) The characteristics of obesity and T2D and a short-term safety evaluation of Akk in humans with obesity. Individuals with obesity and diabetes have increased *Firmicutes* and decreased Akk, accompanied by chronic low-grade inflammation, IR, and elevation of BMI and BW. Two clinical randomized, double-blind, placebo-controlled proof-of-concept and feasibility studies in overweight and obese human volunteers uncovered that oral supplementation with alive or pasteurized 10^9^ and 10^10^ Akk for three months was safe and well tolerated. Alive or pasteurized Akk ameliorates insulin sensitivity, decreases insulinemia and plasma total cholesterol, and slightly reduces body weight. Further plasma metabolome analysis revealed that short-term supplementation with alive or pasteurized Akk significantly modulates amino acid metabolism such as arginine, alanine, tyrosine, phenylalanine, tryptophan, and glutathione metabolism, and increases acylcarnitines and acetoacetate levels via induction of ketogenesis mediated by enhanced β-oxidation. (**B**) The beneficial metabolic effects of Akk on different organs and/or tissues. Akk improves neurodegenerative impairment in the brain and ER stress and inflammation in muscle induced by HFD and/or LPS via the TLR-4 signaling pathway. Akk and its metabolites acetate and propionate regulate the lipid metabolism of the liver, adipose tissue, and organoids and display genotype differences. Alive Akk and EVs protect bone loss induced by Ovx via increasing bone mass, strengthening osteogenic activity, and inhibiting osteoclast formation, whereas pasteurized akk fails to do so. (**C**) Akk and its components impact intestinal metabolism and health. Alive Akk, pasteurized Akk, and outer membrane protein Amuc_1100 activate TLR2/4 to facilitate the secretion of mucins and expression of TJ proteins to promote intestinal barrier function in obesity and T2D induced by WD, HFD, and HFHC. Furthermore, Akk also ameliorates intestinal barrier damage via inhibiting proinflammatory cytokines (including TNF-α, IL-1β, and IL-6) and endotoxin/LPS production. For instance, EVs promote TJ expression by activating AMPK and Amuc_1434; upregulate cell cycle, p53 expression, and TRAIL-Caspase-8 pathway to promote apoptosis; and ultimately suppress cancer cell viability. Akk can increase thermogenesis by interacting with ICAM-2 and its secreted P9 protein stimulates L cells to secrete GLP-1, which maintain glucose homeostasis mediated by IL-6. Akk also suppresses carbohydrate absorption and accelerates energy expenditure, spontaneous physical activity, and intestinal epithelial turnover. The genotypes of Akk impact the efficiency of its inhibiting brown adipose tissue whitening and inflammation. (**D**) The forms of Akk and its outer membrane components. Abbreviations: BMI: body mass index; IR: insulin resistance; BW: body weight; WD: Western diet; HFD: high-fat diet; HFHC: high-fat high cholesterol; ovx: ovariectomy; T1D: type 1 diabetes; T2D: type 2 diabetes; GLP-1: glucagon-like peptide-1; TLRs: Toll-like receptors; ER: endoplasmic reticulum; LPS: lipopolysaccharide; FIAF: fasting-induced adipose factor; GPRs: G protein-coupled receptors; HDACs: histone deacetylases; PPARγ: peroxisome proliferator-activated receptor gamma; CNS: central nervous system; TJ: tight junction.

**Figure 2 ijms-24-03900-f002:**
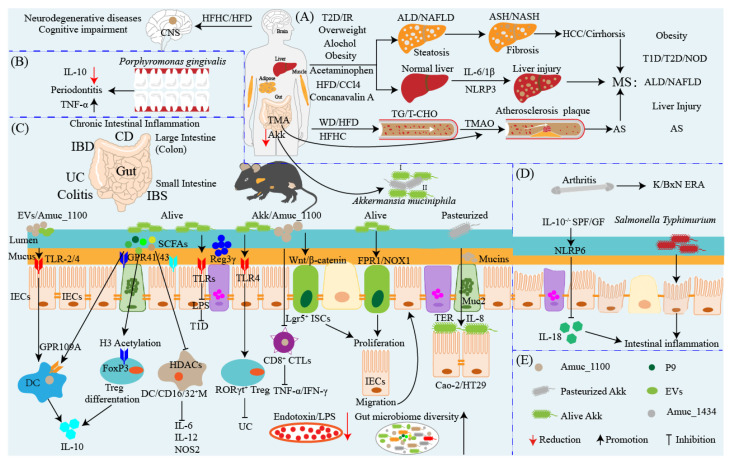
The duality and challenge of *Akkermansia muciniphila* in metabolic syndrome and inflammation therapy. Metabolic syndromes include ALD, NAFLD, NASH, neurodegenerative diseases induced by diets such as HFD/HFHC, and atherosclerosis, and have become one of the major challenges of diseases affecting global public health. Akk as the next promising candidate probiotic shows outstanding antimetabolic syndrome potential. (**A**) Long-term HFD/HFHC diets usually result in neurodegenerative diseases such as cognitive impairment, impaired spatial working memory and novel object recognition, and damaged brain metabolism in early life, while supplementation with Akk can prevent it. T2D, overweightness, obesity, and long-term alcohol consumption generally accompanied by and/or leading to NAFLD/ALD, which include simple steatosis, fibrosis, and cirrhosis, all of which can deteriorate towards ASH/NASH and ultimately deteriorate to HCC. Dietary supplement of Akk ameliorates NAFLD/ALD via regulating lipid and glucose metabolism and intestinal microbiota and eventually improves body metabolic health. Furthermore, HFD usually induces chronic liver injury and inflammatory response, whereas supplementation with Akk saves it by regulating lipid metabolism and inhibiting inflammation. AS is the main contributor to cardiovascular mortality, which is mainly induced by TG/T-CHO, excessive accumulation in WD/HFD/HFHC diets and results in narrowed or blocked blood vessels that drive the cardiovascular events that occur. TMA, one of the metabolites of choline, is metabolized by the gut microbiota to produce TMAO that is proatherogenic and accelerates PROGRESSION, whereas Akk regulates lipid metabolism and inflammatory response to inhibit AS. (**B**,**C**) The anti-inflammation and proinflammatory function of Akk in the gut and other organs. (**B**) Akk ameliorates *porphyromonas-gingivalis*-induced periodontitis by inhibiting bone destruction and TNF-α proinflammatory cytokine secretion and upregulating IL-10 production. (**C**) Akk improves chronic and acute intestinal inflammation including IBD, colitis, IBS, and pathogenic or fungal infection by multiple signaling pathways. For instance, EVs and/or Amuc_1100 activate TLR2/4 to stimulate DCs and secrete anti-inflammatory cytokine IL-10 to repress inflammation. Akk also produces butyrate, which can activate the G protein-coupled receptor 109A (GPR109A) of DCs to promote IL-10 production. Butyrate also directly suppresses the HDACs of DCs/macrophages to inhibit the production of IL-6, IL-22, and NOS2. Propionate, one of the SCFAs derived from Akk, activates GPR43 of Tregs to promote Tregs differentiation to secret IL-10. Akk represses CD8^+^ CTLs to inhibit the production of TNF-α and IFN-γ as well as activates TLRs to inhibit LPS-induced type 1 diabetes (T1D). Akk abundance is decreased and the TLR4 expressing level is upregulated in patients with UC and TLR4 knockout exacerbates intestinal inflammation and is accompanied by the reduction in Akk, whereas supplementation of Akk ameliorates colitis by upregulating RORγt^+^ Treg cell-mediated immune responses. The colonization of Akk is triggered by the interaction of TLR4 and Amuc_1100. Akk not only represses the production and translocation of endotoxin/LPS and recovers the diversity of the microbiota, but also promotes the secretion of antimicrobial peptides and mucins and TJ expression to enhance intestinal barrier integrity. In addition, Akk and Amuc_1100 promote intestinal stem cell proliferation mediated by Wnt/β-catenin and/or FPR1/NOX1 signaling pathways to accelerate intestinal epithelial renewal and protect the gut barrier. Akk enhances transepithelial resistance and inhibits IL-8 production to enhance intestinal epithelial integrity. In addition to intestinal inflammation, gut endotoxin/LPS translocates into the circulatory system and reaches the liver, resulting in liver injury. LPS, acetaminophen, CCl4, and Concanavalin A all can induce acute liver injury, which triggers a severe inflammatory response mediated by NLRP3 inflammasome activation to produce more proinflammatory cytokines IL-6 and IL-1β. (**D**) Akk acts as a pathobiont to promote colitis and exacerbate inflammation progress in some specific conditions. For example, Akk is partly increased in children with enthesitis-related arthritis (ERA), and the administration of the fecal mass of the KRN/B6 × NOD (K/BxN) model of RA mice treated with Akk to human patients with ERA slightly increases ankle swelling and arthritis. Akk exacerbates the inflammatory response caused by murine *Salmonella enterica Typhimurium* and DSS. In both specific pathogen-free and germ-free IL10^-/-^mice, the colonization of Akk mediated by NLRP6 is sufficient for promoting intestinal inflammation by inhibiting IL-18 production modulating the abundance of Akk. (**E**) The forms of Akk and its outer membrane components. Abbreviations: ALD: alcoholic liver disease; ASH: alcoholic steatohepatitis; NAFLD: non-alcoholic fatty liver disease; NASH: non-alcoholic steatohepatitis; HCC: hepatocellular carcinoma; WD: Western diet; HFD: high-fat diet; HFHC: high-fat high-cholesterol; TLRs: Toll-like receptors; LPS: lipopolysaccharide; TG: triglyceride; T-CHO: total cholesterol; AS: atherosclerosis; TMA: trimethylamine; TMAO: trimethylamine-N-oxide; T2D: type 2 diabetes; T1D: type 2 diabetes; NOD: non-obese diabetes; IBD: inflammatory bowel disease; CD: Crohn’s disease; UC: ulcerative colitis; IBS: irritated bowel syndrome; CD8^+^ CTLs: CD8^+^ cytotoxic T lymphocytes; CCl4: carbon tetrachloride; SCFAs: short-chain fatty acids; ERA: enthesitis-related arthritis; K/BxN: KRN/B6xNOD mice; M: macrophages; Tregs: regulatory T cells; DCs: dendritic cells; HDACs: histone deacetylases; TJ: tight junction; MS: metabolic syndromes.

**Figure 3 ijms-24-03900-f003:**
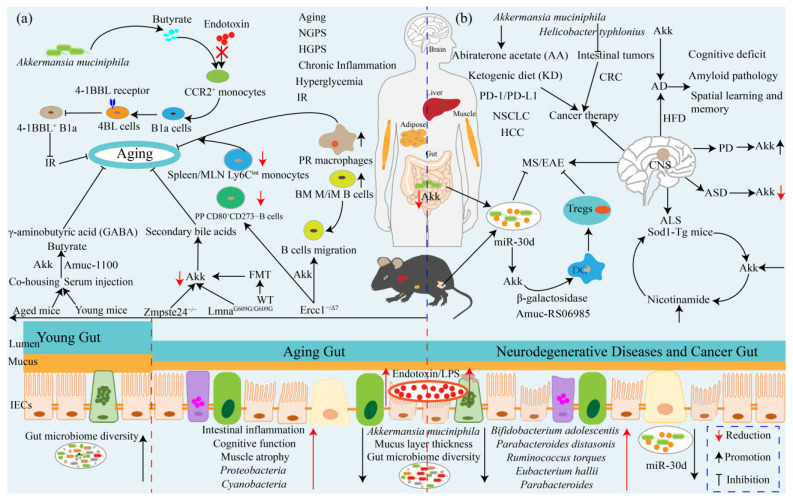
The potential beneficial effects of Akk towards anti-aging, anti-neurodegenerative disease, and anticancer therapy via multiple different signaling pathways mediated by the gut–brain axis. (**a**) Akk and its metabolite butyrate blockade endotoxin-induced CCR2^+^ monocyte activation, which can convert B1a cells into 4BL cells to inhibit 4-1BBL^+^ B1a cell accumulation by activating 4-1BBL receptor and ultimately ameliorate aging-associated IR. Decreased Akk is observed in aged mice, and co-housing with young mice and serum injection from young mice can recover the abundances of Akk and Amuc_1100, which promote butyrate and GABA biosynthesis. In aged mice, administration of Akk improves intestinal function homeostasis and extends the healthy lifespan. In the mouse model of premature aging disorder, Lmna^G609G/G609G^ and Zmpste24^−/−^ mice display decreased *Verrucomicrobia*, and FMT from wild-type mice promotes healthspan and lifespan in both progeroid mouse models; additionally, transplantation with *Verrucomicrobia* Akk is sufficient to exert beneficial effects by restoration of secondary bile acids. Akk ameliorates Ercc1^−/Δ7^mice aging by affecting B cell migration to increase bone marrow mature and immature B cell frequencies and the number of peritoneal resident macrophages while decreasing Peyer’s-patch-activated CD80^+^CD273-B cell frequency and spleen and MLNs Ly6C^int^ monocyte frequencies. Aging gut is usually accompanied by enhanced intestinal inflammation, impaired cognitive function, muscle atrophy, and increased abundance of *Proteobacteria* and *cyanobacteria* and endotoxin and LPS contents while the gut microbiome diversity, mucus layer thickness, and Akk abundance are decreased. (**b**) MicroRNA-30d-5p (miR-30d) promotes β-galactosidase expression and Akk abundance, which in turn promotes Tregs differentiation to suppress MS/EAE symptoms. Akk treatment ameliorates ALS by promoting the production of nicotinamide in the central nervous system of Sod1-Tg mice. The abundance of Akk is decreased in children with autism/ASD, whereas it is increased in patients with PD. The abundance of Akk is decreased in patients and/or mouse models with AD, and Akk treatment ameliorates HFD-induced AD. Akk affects cancer therapy efficacy in multiple cancers. For instance, Akk and *Helicobacter typhlonius* double colonization significantly reduces the number of intestinal tumors, the mucus layer thickness, and goblet cell density of Fabpl Cre; Apc^15lox/+^ mice. In patients with non-small-cell lung cancer (NSCLC) or kidney cancer, Akk relative abundance is associated with clinical responses to ICIs targeting the PD-1/PD-L1 axis, and Akk supplementation restores PD-1 blockade efficacy in non-responder feces FMT mouse. AA treatment increases Akk abundance by uniquely promoting Akk growth. KD can enrich Akk and FMT from the KD gut microbiota, and treatment with Akk confers seizure protection function. In the gut of neurodegenerative diseases and cancer patients, the abundance of *Ruminococcus torque*, *Parabacteroides distasonis*, *Eubacterium hallii*, *Bifidobacterium adolescentis*, and *Parabacteroides*, as well as the endotoxin and LPS content are increased, while the gut microbiome diversity, mucus layer thickness, Akk abundance, and miR-30d level are decreased. Abbreviations: FMT: fecal microbiota transplantation; KD: ketogenic diet; AA: abiraterone acetate; GABA: γ-aminobutyric acid; ASD: autism spectrum disorder; AD: Alzheimer’s disease; PD: Parkinson’s disease; MS: multiple sclerosis; ALS: amyotrophic lateral sclerosis; EAE: experimental autoimmune encephalomyelitis. CRC: colorectal cancer; HCC: hepatocellular carcinoma.

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
