# Peer review of "Akkermansia muciniphila as a Next-Generation Probiotic in Modulating Human Metabolic Homeostasis and Disease Progression: A Role Mediated by Gut–Liver–Brain Axes?"

_ijms, 2023, doi:10.3390/ijms24043900_

Round 1
Reviewer 1 Report
Dear colleagues
I am so fortunate to review this interesting review manuscript entitled with Akkermansia muciniphila as a next-generation probiotic in modulating human metabolic homeostasis and disease progression: a role mediated by gut-liver-brain axes? The paper is well written with novel data about one of an important probiotics. I have only two suggestions; on the possible use of Akkermansia as a prebiotics. Also, include a paragraph about the possible effect of Akkermansia in GIT cancers including the HCC.
Author Response
Dear Reviewers:
Thank you for your letter and for the reviewers’ comments concerning our manuscript entitled “Akkermansia muciniphila as a next-generation probiotic in modulating human metabolic homeostasis and disease progression: a role mediated by gut-liver-brain axes?”. (ID: 2191267). Those comments are all valuable and very helpful for revising and improving our paper, as well as the important guiding significance to our researches. We have studied comments carefully and have made correction which we hope meet with approval. Revised portion are marked in red in the paper. Asides, we also supplemented some new progress that published in recent, please see the point 3. The main corrections in the paper and the responds to the reviewer’s comments are as flowing:
Point 1: 1) on the possible use of Akkermansia as a prebiotics.
Response 1: As discussed in the manuscript, short-term (three months) oral administration of alive or pasteurized Akk is safe for human. Most of current studies demonstrate that Akk has widely biological functions including anti-obesity, anti-diabetes, anti-metabolic syndromes, anti-inflammation, anti-aging, anti-neurodegenerative diseases, and anti-cancer therapy. Thus, Akk as a probiotic, we think it will be used to treat many diseases in the future. In addition to as a probiotic, Akk also have most characteristic and function that is similar to prebiotics, so that Akk maybe use as a prebiotic in the future by dietary intervention that is a conservative but low-risk strategy to increase the abundance of Akk before it is approved for clinical therapy.
Point 2: 2) Also, include a paragraph about the possible effect of Akkermansia in GIT cancers including the HCC.
Response 2: Line 908 We added a paragraph about the possible effect of Akkermansia in GIT cancers including the HCC, accordingly. In addition, we also supplemented new research progress of Akk in cancer, please see lines 961-978.
Lines 909-984: Akk is abundantly present in healthy humans, and recent several reports have found that Akk can influence the efficacy of cancer therapy…………Yet, the above partly reports also imply that Akk may interact with some specific microbes or be affected by the host gut microenvironment, which must be cautious about its application in clinical cancer treatment.
Lines 961-978: Akk is decreased in ovarian cancer patients and closely related to ovarian cancer progression, whereas Akk supplementation with FMT significantly suppresses ovarian cancer progression in mice 155. Akk supplementation with FMT elevate Akk abundance and accompanies acetate accumulation, which is associated with enhanced IFN-γ secretion of CD8+ T cells and its tumor-killing property 155. In colitis-associated colorectal cancer (CAC), Akk abundance is decreased and supplementation with Akk slows down tumourigenesis by expanding CTLs in the colon and MLNs to trigger TNF-α reduction and PD-1 downregulation 100. Fan and colleagues found Akk abundance is significantly reduced in patients with colorectal cancer (CRC) 156. Treatment with Akk suppresses colonic tumorigenesis in ApcMin/+ mice and the growth of implanted HCT116 or CT26 tumors in nude mice by facilitated enrichment of M1-like macrophages in an NLRP3-dependent manner 156. They also found that TLR2 is essential for the activation of the NF-kB/NLRP3 pathway and Akk induced M1-like macrophage response, which the M1-like macrophage and NLRP3/TLR2 are positively associated with Akk in patients with CRC 156. These results emphasize the importance of Akk-induced immune activation in cancer treatment by enhancing gut homeostasis and improving the immune microenvironment and provide a therapeutic target in the GI tract cancers including the HCC.
Point 3: Asides, we also supplemented some new progress that published in recent, please see the flowing:
Lines 555-561: For instance, stable gastrointestinal tract colonization of Akk contribute to the therapy and prognosis of inflammatory intestinal diseases induced by DSS and intestinal radiation via changing host gut microbial community structure and facilitating proliferation and reprogrammed the gene expression profile of Lactobacillus murinus 103. Similarly, cadmium usually induce intestinal damage and treatment with Akk effectively ameliorate intestinal mucosal damage by producing melatonin 104.
Lines 629-637: The relative abundance of Akk is increased in severe fever with thrombocytopenia syndrome virus (SFTSV) infection and reduced in samples from deceased patients 113. Supplementation with Akk could protect against SFTSV infection by suppressing NF-κB-mediated systemic inflammation via producing β-carboline alkaloid harmaline, which can specifically enhance bile acid-CoA: amino acid N-acyltransferase expression in hepatic cells to increase conjugated primary bile acids, glycochenodeoxycholic acid and taurochenodeoxycholic acid 113. Finally, these bile acids induce transmembrane G-protein coupled receptor-5-dependent anti-inflammatory responses to mitigate SFTSV infection 113.
Lines 910-913: As previous mentioned that the relative abundance of Akk is reduced in HCC model mice and continuous treatment with Akk restores the diversity of gut microbiota and reduces liver injury, inflammation, and fibrosis 82.
Lines 961-978: Please see the point 2.
Lines 1075-1083: For example, Akk colonization alleviating high fructose and restraint stress-induced jejunal mucosal barrier disruption by enhancing the function of the NLRP6, promoting autophagy, maintaining the normal secretion of antimicrobial peptides in Paneth cells, promoting the expression of tight junction proteins, negatively regulating the NF-kB signaling pathway and inhibiting the expression of inflammatory cytokines 166. Recent report suggests that Akk upregulates genes involved in maintaining the intestinal barrier function via ADP-heptose-dependent activation of the ALPK1/TIFA pathway, which indicate that Akk promotes intestinal barrier homeostasis by activating innate immune 167.
Lines 1180-1183: You and colleagues found Bacteroides vulgatus SNUG 40005 can restore the abundance of Akk reduction induced by HFD, which enrichment is attributed to metabolites N-acetylglucosamine produced during mucus degradation by Bacteroides vulgatus SNUG 40005 192.
We tried our best to improve the manuscript and made some changes in the manuscript. These changes will not influence the content and framework of the paper. And here we did not list the changes but marked in red in revised paper.
We appreciate for Editors/Reviewers’ warm work earnestly, and hope that the correction will meet with approval.
Once again, thank you very much for your comments and suggestions.
Reviewer 2 Report
Dear authors!
The article submitted for review is devoted to an important topic – Akkermansia muciniphila as a next-generation probiotic in modulating human metabolic homeostasis and disease progression: a role mediated by gut-liver-brain axes?
The review has all the necessary sections and corresponds to the profile, goals and objectives of the journal. The review presents all aspects of the topic under discussion.
I recommend the article for publication in present form.
Author Response
Response to Reviewer 2 Comments
Dear Reviewers:
Thank you for your letter and for the reviewers’ comments concerning our manuscript entitled “Akkermansia muciniphila as a next-generation probiotic in modulating human metabolic homeostasis and disease progression: a role mediated by gut-liver-brain axes?”. (ID: 2191267). Those comments are all valuable and very helpful for revising and improving our paper, as well as the important guiding significance to our researches. We have studied comments carefully and have made correction which we hope meet with approval. Revised portion are marked in red in the paper. Asides, we also supplemented some new progress that published in recent, please see the flowing. The main corrections in the paper and the responds to the reviewer’s comments are as flowing:
Point 1: Asides, we also supplemented some new progress that published in recent, please see the flowing:
Response 1:
Lines 555-561: For instance, stable gastrointestinal tract colonization of Akk contribute to the therapy and prognosis of inflammatory intestinal diseases induced by DSS and intestinal radiation via changing host gut microbial community structure and facilitating proliferation and reprogrammed the gene expression profile of Lactobacillus murinus 103. Similarly, cadmium usually induce intestinal damage and treatment with Akk effectively ameliorate intestinal mucosal damage by producing melatonin 104.
Lines 629-637: The relative abundance of Akk is increased in severe fever with thrombocytopenia syndrome virus (SFTSV) infection and reduced in samples from deceased patients 113. Supplementation with Akk could protect against SFTSV infection by suppressing NF-κB-mediated systemic inflammation via producing β-carboline alkaloid harmaline, which can specifically enhance bile acid-CoA: amino acid N-acyltransferase expression in hepatic cells to increase conjugated primary bile acids, glycochenodeoxycholic acid and taurochenodeoxycholic acid 113. Finally, these bile acids induce transmembrane G-protein coupled receptor-5-dependent anti-inflammatory responses to mitigate SFTSV infection 113.
Lines 910-913: As previous mentioned that the relative abundance of Akk is reduced in HCC model mice and continuous treatment with Akk restores the diversity of gut microbiota and reduces liver injury, inflammation, and fibrosis 82.
Lines 961-978: Akk is decreased in ovarian cancer patients and closely related to ovarian cancer progression, whereas Akk supplementation with FMT significantly suppresses ovarian cancer progression in mice 155. Akk supplementation with FMT elevate Akk abundance and accompanies acetate accumulation, which is associated with enhanced IFN-γ secretion of CD8+ T cells and its tumor-killing property 155. In colitis-associated colorectal cancer (CAC), Akk abundance is decreased and supplementation with Akk slows down tumourigenesis by expanding CTLs in the colon and MLNs to trigger TNF-α reduction and PD-1 downregulation 100. Fan and colleagues found Akk abundance is significantly reduced in patients with colorectal cancer (CRC) 156. Treatment with Akk suppresses colonic tumorigenesis in ApcMin/+ mice and the growth of implanted HCT116 or CT26 tumors in nude mice by facilitated enrichment of M1-like macrophages in an NLRP3-dependent manner 156. They also found that TLR2 is essential for the activation of the NF-kB/NLRP3 pathway and Akk induced M1-like macrophage response, which the M1-like macrophage and NLRP3/TLR2 are positively associated with Akk in patients with CRC 156. These results emphasize the importance of Akk-induced immune activation in cancer treatment by enhancing gut homeostasis and improving the immune microenvironment and provide a therapeutic target in the GI tract cancers including the HCC.
Lines 1075-1083: For example, Akk colonization alleviating high fructose and restraint stress-induced jejunal mucosal barrier disruption by enhancing the function of the NLRP6, promoting autophagy, maintaining the normal secretion of antimicrobial peptides in Paneth cells, promoting the expression of tight junction proteins, negatively regulating the NF-kB signaling pathway and inhibiting the expression of inflammatory cytokines 166. Recent report suggests that Akk upregulates genes involved in maintaining the intestinal barrier function via ADP-heptose-dependent activation of the ALPK1/TIFA pathway, which indicate that Akk promotes intestinal barrier homeostasis by activating innate immune 167.
Lines 1180-1183: You and colleagues found Bacteroides vulgatus SNUG 40005 can restore the abundance of Akk reduction induced by HFD, which enrichment is attributed to metabolites N-acetylglucosamine produced during mucus degradation by Bacteroides vulgatus SNUG 40005 192.
We tried our best to improve the manuscript and made some changes in the manuscript. These changes will not influence the content and framework of the paper. And here we did not list the changes but marked in red in revised paper.
We appreciate for Editors/Reviewers’ warm work earnestly, and hope that the correction will meet with approval.
Once again, thank you very much for your comments and suggestions.